# Item Response Scaling Laws: A Measurement Theory Approach for Efficient and Generalizable Neural Scaling Estimation

**Sang Truong** [* 1]  **Yuheng Tu** [* 2]  **Rylan Schaeffer** [1]  **Sanmi Koyejo** [1]

## Abstract

Scaling laws provide a fundamental framework for understanding the performance of Language Models (LMs), yet deriving them requires prohibitively expensive evaluations across thousands of checkpoints or millions of inference samples. To address this, we introduce Item Response Scaling Laws (IRSL), a unified framework that integrates Item Response Theory (IRT) within the scaling law framework. Unlike traditional approaches that treat each model-benchmark pair in isolation, IRSL disentangles latent model ability from question characteristics, factorizing the scaling law estimation for $M$ models and $N$ questions to significantly reduce parameter complexity from $O(M \times N)$ to $O(M + N)$. We instantiate IRSL with Beta-IRT, which leverages the empirical probability responses of LMs—such as token probabilities in pre-training and pass rates in test-time sampling—to capture richer signals than binary responses. We validate our approach across two prevalent scaling paradigms: (1) pre-training downstream scaling, using 6,612 LM checkpoints and 37,682 questions from 10 benchmarks; and (2) test-time scaling, using 12 LMs and 120 questions from 4 benchmarks with up to 2,500 samples per question. Given a one-time calibration on existing model responses, IRSL yields more reliable scaling estimates using only 50 questions per benchmark (a 99.9% reduction), achieving comparable or superior decision accuracy to traditional approaches. Furthermore, we show that the estimated latent model abilities are generalizable, enabling accurate performance forecasting across benchmarks that share the same measurement objective.

[1]Stanford University [2]University of California, Los Angeles. Correspondence to: Sang Truong <sttruong@cs.stanford.edu>.

*Proceedings of the 43rd International Conference on Machine Learning*, Seoul, South Korea. PMLR 306, 2026. Copyright 2026 by the author(s).

## 1. Introduction

Scaling laws provide a principled framework for predicting performance and allocating resources in Language Models (LMs). We focus on two primary forms: pre-training downstream scaling, which characterizes how performance on downstream tasks improves with pre-training compute (Kaplan et al., 2020; Hoffmann et al., 2022; Biderman et al., 2023; Grattafiori et al., 2024), and test-time scaling, which describes how performance improves with the number of independent inference samples. Test-time scaling encompasses diverse strategies including chain-of-thought prompting, tree-of-thought search, repeated sampling, and reinforcement learning-based reasoning (Brown et al., 2024; Hughes et al., 2024; Levi, 2024); in this work, we focus specifically on the repeated sampling paradigm.

Deriving these laws is computationally expensive. A pre-training scaling study typically requires evaluating thousands of model checkpoints across tens of thousands of questions. Similarly, establishing test-time scaling laws requires a massive number of queries: number of models × number of questions × number of samples per question (typically $10^2 \times 10^5 \times 10^4$). Consequently, practical studies are often constrained to small experimental scales (Chen et al., 2024; Brown et al., 2024; 2020). The laws derived from such limited scales can exhibit unintuitive behaviors. For example, Brown et al. (2024) empirically find a power-law test-time scaling relationship that, as Schaeffer et al. (2025) demonstrates, holds only for specific, ill-structured distributions of single-sample success rates.

To address the cost of evaluation, we turn to Item Response Theory (IRT). Originating in psychology and human testing, IRT is a probabilistic framework that models the interaction between test takers and questions, known for significantly reducing the number of queries required to reliably estimate the ability of test takers. It has been highly successful in both human testing (Lord, 1980) and recent LM leaderboard evaluations (Truong et al., 2025; Hofmann et al., 2025; Kipnis et al., 2025). Building on this, we introduce Item Response Scaling Laws (IRSL), a methodology that integrates IRT into the scaling law framework. IRSL leverages the property of IRT to disentangle the ability of LMs from the characteristics of the questions, factorizing

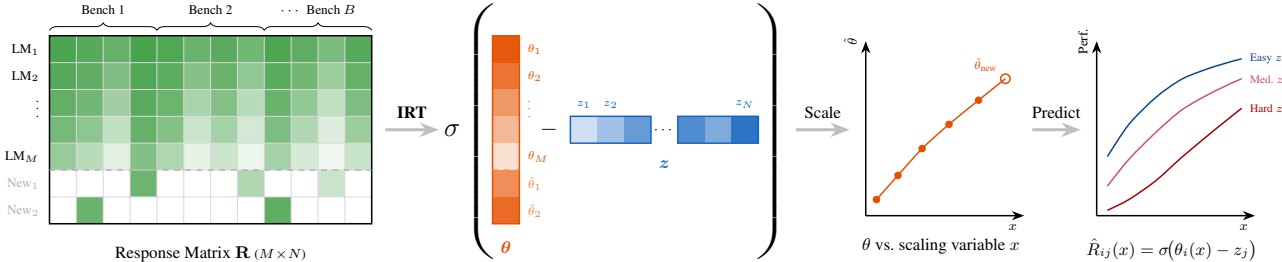

*Figure 1.* **IRSL reduces scaling law estimation from $O(M \times N)$ to $O(M+N)$ by factorizing model ability from question difficulty.** *Left:* The response matrix $\mathbf{R}$ records empirical probabilities across LMs and benchmark questions; sparse rows for new LMs illustrate query efficiency via adaptive testing. *Center-left:* IRT decomposes $\mathbf{R}$ into LM abilities $\boldsymbol{\theta}$ (orange) and question difficulties $\boldsymbol{z}$ (blue), so that $R_{ij} \approx \sigma(\theta_i - z_j)$. *Center-right:* The estimated $\theta$ values scale predictably with the scaling variable $x$ (e.g., pre-training compute or test-time samples). *Right:* Recomposing $\theta(x)$ with the calibrated $z$ yields per-question scaling predictions $\hat{R}_{ij}(x) = \sigma(\theta_i(x) - z_j)$, where questions of varying difficulty produce distinct curves.

the problem into $M$ sets of LM-specific parameters and $N$ sets of question-specific parameters, reducing the complexity from $O(M \times N)$ to $O(M + N)$. This factorization allows the estimated ability to be transferred across benchmarks that share the same measurement objective.

Prior applications of IRT typically rely on binary responses[1]. However, unlike human testing, LMs provide empirical probability responses. In pre-training, LMs yield token probabilities that offer smoother scaling signals than discrete accuracy (Schaeffer et al., 2024; Magnusson et al., 2025). In test-time sampling, LMs provide per-attempt success rates averaged from many independent inferences. Such empirical probability responses convey richer information than binary responses. To leverage this information, we instantiate IRSL with Beta-IRT, which uses a Beta loss to model these empirical probability responses. While IRSL is a general framework compatible with any IRT model, Beta-IRT enables it to exploit the richer probability signals that LMs uniquely provide.

Our contributions are as follows:

- We conduct a large-scale study on 6,612 LM checkpoints and 37,682 questions from 10 benchmarks to demonstrate the effectiveness of our pre-training downstream IRSL. We show that it yields generalizable and robust estimates of scaling behavior with limited query budgets.

- On 12 LMs across 120 questions from 4 benchmarks with up to 2,500 samples per question, preliminary evidence suggests that IRSL similarly applies to test-time scaling.

By embedding the scaling law within the IRT framework, instantiated here via Beta-IRT, our approach provides a theoretically principled and empirically vali-

dated alternative to traditional aggregate performance scaling. Our code is released at `https://github.com/aims-foundations/irsl`.

## 2. Related Work

**Pre-training Loss Scaling Laws** Many neural networks exhibit power-law scaling for the pre-training loss as a function of compute, data, or parameters (Hestness et al., 2017; Kaplan et al., 2020; Bahri et al., 2021; Hernandez et al., 2021; Hoffmann et al., 2022; Muennighoff et al., 2024; Brown et al., 2020).

**Downstream Performance Scaling Laws** Unlike predicting loss, predicting downstream performance from scale is generally harder (Lourie et al., 2025; Schaeffer et al., 2024). However, recent work has demonstrated that it can be done based on a two-step prediction that chains together predictions from scale to loss and loss to downstream performance (Biderman et al., 2023; Magnusson et al., 2025; Gadre et al., 2024).

**Test-time Scaling Law** Test-time scaling laws characterize how a model's performance on a benchmark (e.g., success rate) improves as the number of stochastic samples drawn at inference increases, typically following a power law (Brown et al., 2024; Snell et al., 2024; Hughes et al., 2024). Later works demonstrate that such a power relationship holds only for ill-structured response distributions in single-sample success rates (Schaeffer et al., 2025; Levi, 2024)

**Efficient LM Evaluation** Several recent works adopt Item Response Theory (IRT) as a foundation for LM evaluation using binary responses and Bernoulli loss, which we refer to as Binary-IRT. (Truong et al., 2025; Hofmann et al., 2025; Kipnis et al., 2025; Polo et al., 2024). Binary-IRT has been shown to outperform many efficient evaluation methods, such as Anchor Points (Vivek et al., 2024),

---
[1] Where the response of a test taker to a question is either correct or incorrect.

SMART (Gupta et al., 2025), MAGI (Paech, 2024), and Stratified Sampling (Perlitz et al., 2024). Our contribution integrates this framework into the scaling law estimation scenario and further uses Beta-IRT, which leverages empirical probability responses unique to LMs to achieve better performance than Binary-IRT.

**Continuous IRT Models** Traditional IRT relies on binary responses. Chen et al. (2019) propose $\beta^3$-IRT, which uses a three-parameter Beta distribution to model continuous responses. Our Beta-IRT differs in that we parameterize the Beta distribution mean via the standard IRT logistic function $\sigma(d(\theta-z))$, preserving the interpretability of ability $\theta$ and difficulty $z$ while coupling naturally with scaling law estimation. The key novelty of IRSL is not IRT for evaluation *per se*, but the integration of IRT into the scaling law framework for prediction.

## 3. Method

Item Response Theory (IRT) provides an elegant mathematical framework to model the interaction of LMs and benchmark questions. We show how, under this framework, various known scaling laws arise naturally, and how the framework facilitates efficient and generalizable scaling laws estimation. We show the definitions, traditional fitting approaches, and IRT-based fitting approaches of the scaling laws in Table 1.

### 3.1. Traditional Binary-IRT

Evaluating $M$ models on benchmarks with $N$ questions requires $M \times N$ queries, which is prohibitively expensive at scale. Item Response Theory addresses this by modeling the interaction between test taker ability and question difficulty, enabling reliable evaluation from far fewer queries. Formally, IRT refers to a class of probabilistic latent variable models that explain the relationship between the test taker's latent ability, the question's characteristics (e.g., difficulty), and the observed response from the test taker to the questions (Baker, 2001; Van der Linden et al., 2000). A central model in IRT is the 1PL model (Rasch, 1993), where each test taker has an ability parameter $\theta$, and each question has a difficulty parameter $z$. A higher $\theta$ denotes greater ability, and a higher $z$ denotes a more difficult question. Let $y$ denote the binary response of the test taker to the question, where $y = 1$ if the response is correct and 0 otherwise. The probability of a correct response is modeled by $p(y = 1 \mid \theta, z) = \sigma(\theta - z)$, where $\sigma$ is the sigmoid function. Another widely adopted model in IRT is the 2PL model (Lord, 1952; Birnbaum, 1968), which adds a discrimination parameter $d$ to capture how sharply a question differentiates between test takers of different abilities, modeling the probability of a correct response as $p(y = 1 \mid \theta, z, d) = \sigma(d \cdot (\theta - z))$. The difficulty $z$ and the discrimination $d$ are collectively referred to as the item parameters. The use of IRT consists of two phases: calibration, which estimates the item parameters, and adaptive testing, which enables efficient ability estimation for new test takers.

During calibration, a binary response matrix $Y$ of size $M \times N$ is collected, where $M$ and $N$ denote the number of test takers and questions, respectively. Entry $Y_{ij}$ represents the response of test taker $i$ to question $j$. With the binary response matrix, the item parameters can be estimated via either MLE or EM by minimizing the Bernoulli loss between the IRT predicted probabilities and the observed binary responses $\mathcal{L}_{\text{Bernoulli}} = -\sum_{i=1}^{M} \sum_{j=1}^{N} [Y_{ij} \log p_{ij} + (1 - Y_{ij}) \log(1 - p_{ij})]$, where $p_{ij} = \sigma(d_j(\theta_i - z_j))$ for the 2PL model (or $p_{ij} = \sigma(\theta_i - z_j)$ for 1PL) (Bock & Aitkin, 1981; Chalmers, 2012; Wu et al., 2020).

During adaptive testing, the ability of a new test taker is efficiently estimated through an iterative procedure that alternates between ability update and question selection. In the ability update step, the test taker's ability is estimated from their responses to all previously asked questions. In the question selection step, the most informative question is selected for the query based on the current ability estimate. Consequently, significantly fewer questions are required to obtain a reliable estimate of the new test taker's ability (e.g., 50 out of 37,682 in our experiments; see Section 4.2) (Meijer & Nering, 1999; Chang, 2015).

### 3.2. Traditional Scaling Laws

We investigate two scaling laws: the pre-training downstream scaling law and the test-time scaling law. The pre-training downstream scaling law characterizes how the performance of an LM $i$ on a benchmark $\mathcal{D}$ scales with the pre-training compute FLOP. The traditional approach involves a two-step fitting process: first modeling the relationship between pre-training loss $L$ and compute FLOP, and subsequently mapping the loss $L$ to benchmark performance $\text{Performance}(i, \mathcal{D})$ (Bhagia et al., 2024):

$$L \approx \alpha \cdot \text{FLOP}^{-\beta} + \gamma,$$
$$\text{Performance}(i, \mathcal{D}) \approx a \cdot \sigma(b \cdot (L - l_0)) + c, \qquad (1)$$

where $\sigma$ denotes the sigmoid function, and $\alpha, \beta, \gamma, a, b, c,$ and $l_0$ are learnable parameters. The sigmoid mapping in the second step is known to be sensitive to initialization and hyperparameters; our IRT-based approach (Section 3.4) avoids this two-step fitting. Following Bhagia et al. (2024), we use the benchmark-specific loss[2] as $L$. Consequently, all scaling law parameters are benchmark-

---

[2] Can be understood as the pre-training validation loss on benchmark questions.

| | Definition | Traditional Fitting Approach | IRT-based Fitting Approach |
|---|---|---|---|
| **Pre-training** Acc | $\mathrm{Acc}(i,\mathcal{D}) = \frac{1}{N}\sum_{j=1}^{N} Y_{ij}$ | $a \cdot \sigma(b \cdot (\alpha \cdot \mathrm{FLOP}^{-\beta} + \gamma - l_0)) + c$ | $\frac{1}{N}\sum_{j=1}^{N} \sigma(d_j \cdot (a \cdot \log(\mathrm{FLOP}_i) + b - z_j))$ |
| **Pre-training** p$_{\text{Correct Choice}}$ | $\mathrm{p}_{\text{Correct Choice}}(i,\mathcal{D}) = \frac{1}{N}\sum_{j=1}^{N} \mathrm{p}_{\text{Correct Choice}}(i,j)$ | $a \cdot \sigma(b \cdot (\alpha \cdot \mathrm{FLOP}^{-\beta} + \gamma - l_0)) + c$ | $\frac{1}{N}\sum_{j=1}^{N} \sigma(d_j \cdot (a \cdot \log(\mathrm{FLOP}_i) + b - z_j))$ |
| **Test-time** pass@k | $\mathrm{pass@k}(i,\mathcal{D}) = \frac{1}{N}\sum_{j=1}^{N} \mathrm{pass@k}(i,j)$ | $\frac{1}{N}\sum_{j=1}^{N}(1 - (1 - \mathrm{pass@1}(i,j))^k)$ | $\frac{1}{N}\sum_{j=1}^{N}(1 - (1 - \sigma(d_j \cdot (\theta_i - z_j)))^k)$ |

*Table 1.* **IRSL learns question-level parameters, enabling generalization across question sets with the same measurement objective.** Definitions, traditional fitting approach, and IRT-based fitting approach for Acc, p$_{\text{Correct Choice}}$ (pre-training downstream scaling law), and pass@k (test-time scaling law), using the 2PL model. Traditional approaches fit parameters specific to LMs and benchmarks.

and LM-specific, implying that parameters derived for one LM-benchmark pair do not generalize to others. $\mathrm{Performance}(i,\mathcal{D})$ can be quantified using metrics such as accuracy (Acc) or the average probability of the correct choice (p$_{\text{Correct Choice}}$). Previous work notes that discrete metrics like Acc can exhibit performance jumps across scales, whereas continuous metrics like p$_{\text{Correct Choice}}$ often reveal more predictable improvements (Schaeffer et al., 2024; Magnusson et al., 2025). See Appendix A for the calculation details of $L$, Acc, and p$_{\text{Correct Choice}}$.

The test-time scaling law characterizes the relationship between the success rate of an LM $i$ on a benchmark $\mathcal{D}$ and the number of independent inference samples $k$ (Brown et al., 2024; Levi, 2024). For an LM $i$ and a question $j$, $\mathrm{pass@1}(i,j)$ is defined as the probability that a single sample from LM $i$ correctly answers question $j$. The question-level success rate, $\mathrm{pass@k}(i,j)$, is defined as the probability that at least one of the $k$ generated responses is correct. The benchmark-level success rate $\mathrm{pass@k}(i,\mathcal{D})$ is computed by averaging the probabilities over all benchmark questions $\mathrm{pass@k}(i,\mathcal{D}) = \frac{1}{N}\sum_{j=1}^{N} \mathrm{pass@k}(i,j)$. Previous studies empirically find that $-\log \mathrm{pass@k}$ exhibits a power-law decay with respect to $k$ (Brown et al., 2024; Hughes et al., 2024): $-\log \mathrm{pass@k}(i,\mathcal{D}) \approx uk^{-v}$, where $u$ and $v$ are scaling law parameters. Schaeffer et al. (2025) note that while the question-level success rate theoretically scales exponentially with $k$, the benchmark-level power law emerges because the distribution of $\mathrm{pass@1}(i,j)$ is heavy-tailed towards extremely difficult questions. The relationship between $\mathrm{pass@k}(i,\mathcal{D})$ and $\mathrm{pass@1}(i,j)$ can be expressed as:

$$\mathrm{pass@k}(i,\mathcal{D}) = \frac{1}{N}\sum_{j=1}^{N}(1 - (1 - \mathrm{pass@1}(i,j))^k), \quad (2)$$

where pass@1 is benchmark- and LM-specific.

### 3.3. Beta-IRT
Unlike human testing, LMs provide empirical probability responses that convey richer information than binary responses, such as p$_{\text{Correct Choice}}$ from pre-training downstream scaling and pass@1 from test-time scaling. Drawing on insights from Beta regression (Ferrari & Cribari-

Neto, 2004) and related continuous IRT models (Chen et al., 2019), we propose Beta-IRT, which replaces the standard Bernoulli loss with the Beta loss: $\mathcal{L}_{\text{Beta}} = -\sum_{i=1}^{M}\sum_{j=1}^{N} \log p(P_{ij}; p_{ij}, \phi)$, where $P_{ij}$ denotes the empirical response probability, $p_{ij}$ denotes IRT predicted probability, and $\phi > 0$ is a precision parameter controlling the concentration of the Beta distribution around its mean (higher $\phi$ yields a tighter distribution). Unlike $\beta^3$-IRT (Chen et al., 2019), which uses a three-parameter Beta distribution, our formulation parameterizes the Beta mean via the standard IRT logistic function, preserving the interpretability of $\theta$ and $z$. We empirically find that Beta-IRT achieves reliable calibration with significantly fewer test takers than Binary-IRT, substantially reducing calibration costs.

### 3.4. Item Response Scaling Laws
The core idea is to model p$_{\text{Correct Choice}}$ and pass@1 within the IRT framework. For the pre-training downstream scaling law, we employ a two-stage fitting procedure: first mapping pre-training compute FLOP to the ability $\theta$, and subsequently mapping $\theta$ to the benchmark performance $\mathrm{Performance}(i,\mathcal{D})$. Empirically, we observe that the $\theta$ scales linearly with $\log \mathrm{FLOP}$ (Figure 12):

$$\theta_i \approx a \cdot \log(\mathrm{FLOP}_i) + b,$$

$$\mathrm{Performance}(i,\mathcal{D}) \approx \frac{1}{N}\sum_{j=1}^{N} \sigma(d_j \cdot (\theta_i - z_j)), \quad (3)$$

where $a$, $b$, and $\theta_i$ are LM-specific parameters, and $d_j$ and $z_j$ are question-specific parameters. Specifically, for the baseline scenario where $\mathrm{Performance}(i,\mathcal{D})$ is measured by accuracy, we employ Binary-IRT with binary responses. For our approach, where $\mathrm{Performance}(i,\mathcal{D})$ is measured by p$_{\text{Correct Choice}}$, we employ Beta-IRT with empirical probability responses.

With calibrated item parameters, adaptive testing enables the efficient estimation of a new LM's ability using fewer questions, facilitating the rapid derivation of its pre-training downstream scaling law. Furthermore, IRSL offers generalizability across benchmarks. For a target benchmark $\mathcal{D}'$ sharing the same measurement objective as $\mathcal{D}$, the $\theta$ es-

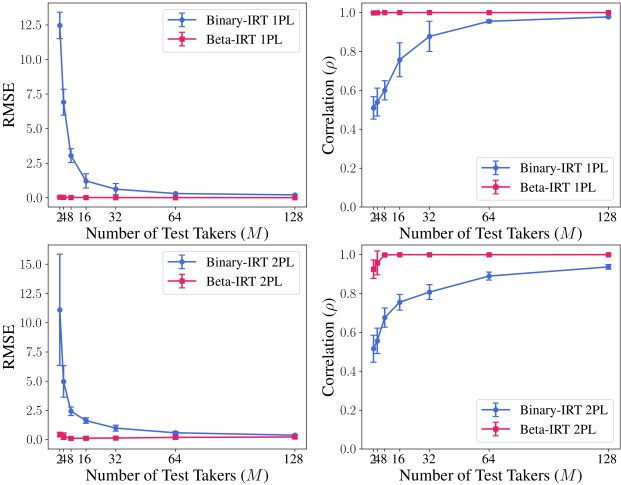

*Figure 2.* **Beta-IRT achieves reliable calibration with as few as 2 test takers, requiring 30–60× fewer than Binary-IRT.** We report RMSE (Left) and Correlation (Right) for both the 1PL model (Top) and the 2PL model (Bottom) as a function of the number of test takers $M$. Error bars indicate $\pm 1$ standard deviation over 10 trials.

timated from $\mathcal{D}$ is transferable. This allows for the prediction of performance on $\mathcal{D}'$ via $\text{Performance}(i, \mathcal{D}') \approx \frac{1}{N'} \sum_{j=1}^{N'} \sigma(d'_j \cdot (\theta_i - z'_j))$, obviating the need to collect empirical responses from LM $i$ on $\mathcal{D}'$.

For the test-time scaling law, we model the benchmark-level success rate by substituting the Beta-IRT predicted single-attempt probability for $\text{pass@}1(i, j)$:

$$\text{pass@k}(i, \mathcal{D}) = \frac{1}{N} \sum_{j=1}^{N} (1 - (1 - \sigma(d_j \cdot (\theta_i - z_j)))^k), \quad (4)$$

where $\theta_i$ is an LM-specific ability parameter estimated per benchmark, and $d_j$ and $z_j$ are question-specific parameters. Similar to pre-training downstream scaling, our approach enables efficient estimation of a new LM's ability using fewer questions, and the ability can generalize across different benchmarks sharing the measurement objective. Furthermore, in test-time scaling, a binary response tensor of shape $M \times N \times K$ is first collected, where $K$ denotes the total number of samples. This tensor is averaged across the sample dimension to yield an empirical probability response matrix. In this setting, we empirically find that Beta-IRT facilitates the efficient estimation of a new LM's ability using significantly fewer samples, further enhancing query efficiency.

## 4. Experiments

In Section 4.1, we conduct a simulation study to demonstrate the superior sample efficiency of Beta-IRT. In Section 4.2, we demonstrate the advantages of the Item Response Scaling Law (IRSL) for pre-training downstream

scaling, and in Section 4.3, we preliminarily validate its effectiveness for test-time scaling.

### 4.1. Sample Efficiency of Beta-IRT

To quantify the information gain provided by empirical response probabilities, we conduct controlled simulations comparing the standard Binary-IRT with our proposed Beta-IRT for both 1PL and 2PL models. We generate true abilities $\theta_i \sim \mathcal{N}(0, 1)$ for $M$ test takers and question difficulties $z_j \sim \mathcal{N}(0, 1)$ for $N = 100$ questions. For the 2PL model, question discriminations are sampled from $d_j \sim \text{LogNormal}(0, 0.5)$. We simulate binary response matrices $Y_{ij} \sim \text{Bernoulli}(p_{ij})$ and empirical probability matrices $P_{ij} = p_{ij} + \varepsilon_{ij}$, where the noise term $\varepsilon_{ij} \sim \mathcal{N}(0, 0.01^2)$ mimics empirical uncertainty.

We vary the number of test takers $M$ across the set $\{2, 4, 8, 16, 32, 64, 128\}$, a range chosen to reflect the typical availability of test takers in LM evaluation. We report the Root Mean Square Error (RMSE) and Pearson correlation coefficient ($\rho$) between the estimated and true item parameters, averaged over 10 independent trials. Figure 2 illustrates the substantial sample efficiency advantage of Beta-IRT. In the 1PL setting, Beta-IRT achieves near-perfect parameter recovery (RMSE $< 0.05$, $\rho > 0.999$) with as few as $M = 2$ test takers. In contrast, Binary-IRT requires significantly larger sample sizes ($M \geq 64$) to attain comparable accuracy. The 2PL model exhibits a similar trend: Beta-IRT 2PL maintains an RMSE $< 0.7$ across all sample sizes, while Binary-IRT 2PL begins with a high error and only approaches the performance of Beta-IRT 2PL at $M = 128$. These findings confirm that Beta-IRT significantly improves sample efficiency for calibration, reducing the high computational costs associated with large-scale LM benchmarking.

### 4.2. Pre-training Downstream IRSL

We use the data suite from DataDecide (Magnusson et al., 2025), a large-scale controlled experiment on pre-training downstream scaling. The objective is to identify which of the 25 pre-training data mixtures yields the highest benchmark accuracy for the target model size (here, 1B). Because LMs are expensive to pretrain, standard practice involves fitting scaling laws on smaller models and extrapolating to the target size. The suite comprises models pre-trained on 25 data mixtures across 14 model sizes, ranging from 4M to 1B parameters. Each run includes 6 to 30 checkpoints depending on the model size, resulting in a total of 6,612 model checkpoints. All checkpoints are evaluated on 10 multiple-choice benchmarks, totaling 37,682 questions. From this, we extract two response matrices of shape $6612 \times 37682$: a binary response matrix and an empirical probability response matrix. We randomly select 5 data mixtures to serve as the train set for calibration. The re-

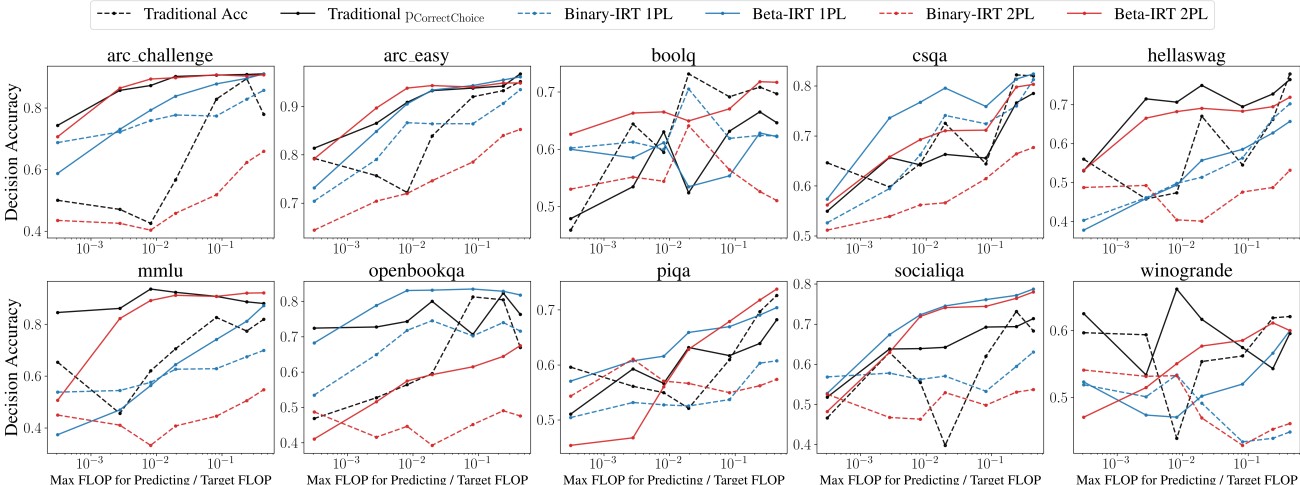

*Figure 3.* **Beta-IRT provides more robust scaling law estimates, especially on lower-quality benchmarks.** Decision Accuracy vs. Proportion of Target FLOPs across 10 benchmarks. We iteratively fit scaling laws by including larger models and extrapolating to the target size to predict benchmark accuracy rankings. Results are averaged over five random train-test splits. Black lines denote Traditional Scaling; Blue and Red lines denote IRSL 1PL and 2PL, respectively. Dashed lines indicate binary responses (Acc), while solid lines indicate empirical probability responses ($p_{\text{Correct Choice}}$).

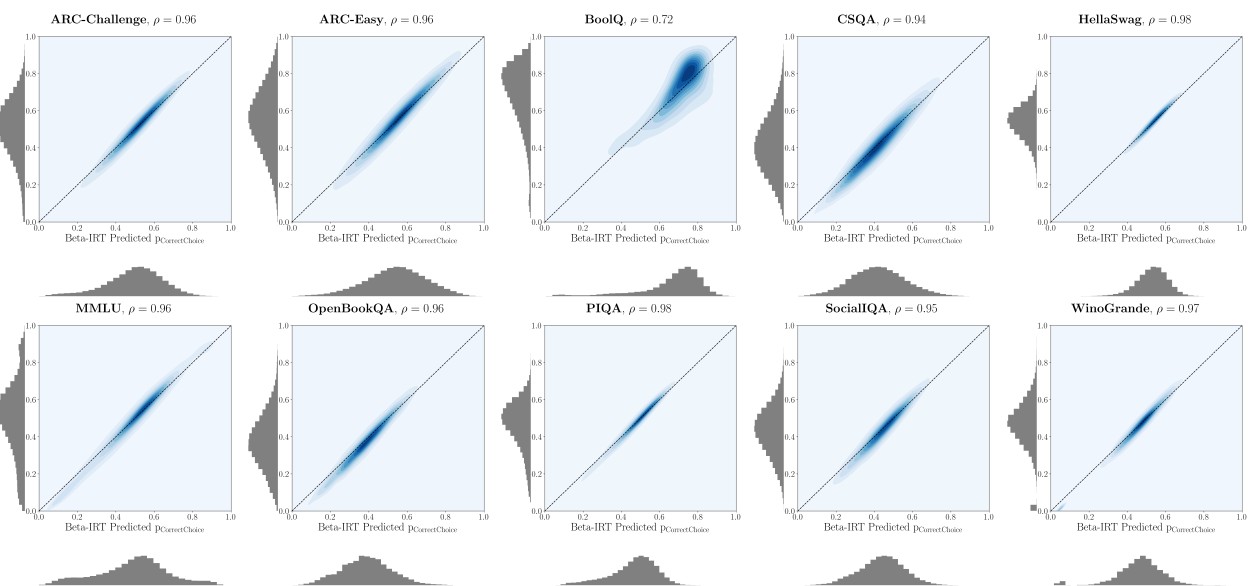

*Figure 4.* **Beta-IRT effectively captures the underlying response structure across all 10 benchmarks.** Correlation between Beta-IRT 2PL predicted $p_{\text{Correct Choice}}$ (x-axis) and empirical $p_{\text{Correct Choice}}$ (y-axis), visualized using 2-D KDE contour plots. The Pearson correlation coefficient ($\rho$) is reported for each benchmark, with marginal histograms showing the $p_{\text{Correct Choice}}$ distribution. The corresponding results for 1PL are provided in Figure 16 in Appendix B.

maining 20 data mixtures constitute the test set for adaptive testing, where we estimate the ability $\theta$ using a budget of only 50 questions per benchmark.

We evaluate the effectiveness of a scaling law method using Decision Accuracy, a metric that quantifies rank consistency. Let $\mathcal{P}$ denote the set of all pairs of data mixtures $(A, B)$ in the test set. Let $y$ and $\hat{y}$ represent the ground truth benchmark accuracy at the 1B target size and the predicted performance extrapolated from the scaling law, re-

spectively. Decision Accuracy is defined as:

$$\text{Decision Accuracy} =$$
$$\frac{1}{|\mathcal{P}|} \sum_{(A,B) \in \mathcal{P}} \mathbb{I}(\text{sign}(\hat{y}_A - \hat{y}_B) = \text{sign}(y_A - y_B)). \quad (5)$$

We iteratively include larger models for the scaling law fitting and extrapolate to the target size to predict the benchmark accuracy rankings. Figure 3 reports the Decision Ac-

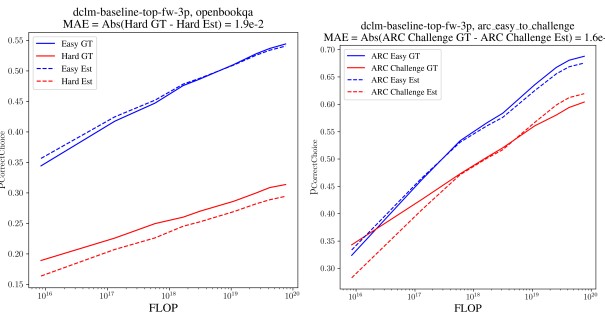

*Figure 5.* **IRSL accurately predicts scaling trends on harder sets using the ability estimated from easy sets alone.** (Left) Within-benchmark transfer on OpenBookQA. (Right) Cross-benchmark transfer from ARC Easy to ARC Challenge. Solid lines represent the Ground Truth (GT) scaling curves, while dashed lines represent the estimated curves where LM ability is derived solely from the easy set.

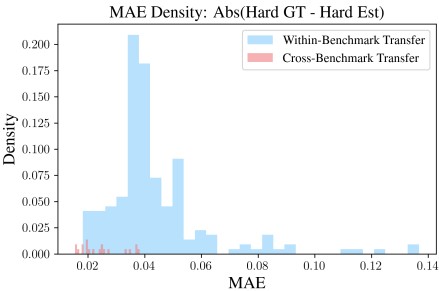

*Figure 6.* **The ability $\theta$ estimated by IRSL is robustly transferable across benchmark sets.** MAE distribution for hard set estimation across all benchmarks and LM data mixtures. We report the MAE between the ground truth scaling curve and the estimated curve for two settings: Within-Benchmark Transfer (blue) and Cross-Benchmark Transfer (red). See Figure 19 for the full results.

curacy against the proportion of target FLOPs across 10 benchmarks. We compare six scaling law methods: traditional scaling law (using $\mathrm{Acc}$ or $p_{\mathrm{Correct\ Choice}}$ via Equation 1) and IRSL (Binary-IRT and Beta-IRT, using 1PL and 2PL variants via Equation 3). On ARC Challenge, ARC Easy, and MMLU, Beta-IRT matches the strong performance of Traditional $p_{\mathrm{Correct\ Choice}}$, and they outperform other methods. On CommonsenseQA, OpenBookQA, PIQA, and SocialIQA, Beta-IRT demonstrates superior reliability, outperforming other methods. On BoolQ, HellaSwag, and WinoGrande, Beta-IRT fails to capture a predictive trend. We attribute this to benchmark homogeneity: the calibrated item parameters on these benchmarks are highly concentrated (e.g., BoolQ: $\sigma_z{=}0.19$, $\sigma_d{=}0.14$), meaning nearly all questions have similar difficulty and discrimination. As a result, the Test Information Function (TIF) per item is low, limiting IRT's ability to differentiate model abilities—in contrast to benchmarks like ARC Challenge ($\sigma_z{=}0.55$, $\sigma_d{=}0.61$), where diverse items provide substantially more information (see Figure 20 in Ap-

pendix C). These observations align with the findings of Heineman et al. (2025), a follow-up study on DataDecide that introduces a signal-to-noise ratio to assess benchmark quality in downstream scaling. Specifically, we find that Beta-IRT ties with Traditional $p_{\mathrm{Correct\ Choice}}$ on high-quality benchmarks, outperforms other methods on lower-quality benchmarks, and fails to capture a trend on extremely noisy benchmarks. We conclude that Beta-IRT provides a more robust estimate of the scaling law curve with limited query budget, especially for benchmarks with lower quality. We report the scaling curve fitting for the six methods in Figure 13, 14, and 15 in Appendix B.

We report the strong correlation between Beta-IRT predicted $p_{\mathrm{Correct\ Choice}}$ and the empirical $p_{\mathrm{Correct\ Choice}}$ on the test set, as illustrated in Figure 4 for the 2PL variant and Figure 16 for the 1PL variant. We conclude that Beta-IRT effectively captures the underlying response structure. We further report the Beta-IRT curve on single questions in Figure 17 and 18 in Appendix B.

Next, we demonstrate the generalizability of IRSL across benchmark sets with different difficulties. We partition each benchmark into an easy and a hard subset based on the mean $p_{\mathrm{Correct\ Choice}}$ of each question across all LM checkpoints. We estimate the ability $\theta$ of each LM checkpoint using only the easy subset. Then, using these $\theta$ estimates alongside the calibrated item parameters of the hard subset, we generate the scaling curve for the hard subset without accessing the responses. Figure 5 (Left) illustrates this within-benchmark transfer for OpenBookQA on a representative LM data mixture. We further demonstrate cross-benchmark transfer in Figure 5 (Right), showing that $\theta$ estimated on ARC Easy effectively predicts the scaling curve on ARC Challenge. Figure 6 reports the distribution of Mean Absolute Error (MAE) between the ground truth and the estimated scaling curve on the hard sets across all benchmarks and data mixtures (full results in Figure 19). We conclude that the ability estimated by IRSL is transferable, enabling reliable performance forecasting on benchmark sets with the same measurement objective. In Appendix D, we further discuss how to assess similar measurement objectives and show that cross-benchmark transfer may be more broadly applicable.

### 4.3. Test-time IRSL

We collect a binary response tensor of shape $12 \times 120 \times 2500$ (12 LMs, 120 questions from 4 benchmarks, 2500 samples, listed in Appendix E). We obtain the empirical $\mathrm{pass@1}$ response matrix by averaging over the last dimension. We filter out questions with extremely low $\mathrm{pass@1}$ as they offer no discriminatory power. In each train-test split, we randomly select 8 LMs to serve as the training set for calibration, while the remaining 4 LMs constitute the test set for adaptive testing with a query budget of 50 samples

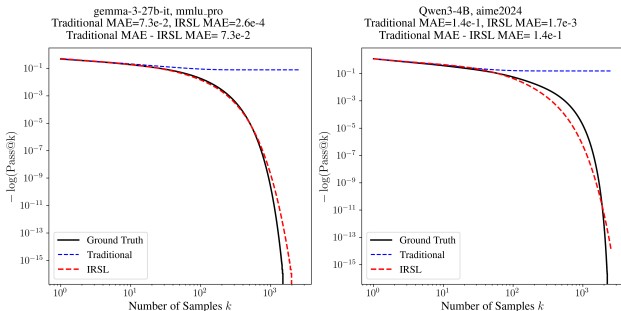

*Figure 7.* **IRSL yields more reliable test-time scaling estimates than traditional approaches given a limited query budget.** Comparison of three test-time scaling curves: Ground Truth, Traditional scaling law, and IRSL, for two representative LM-Benchmark pairs in the test set. We plot $-\log \text{pass@k}$ against the number of samples $k$.

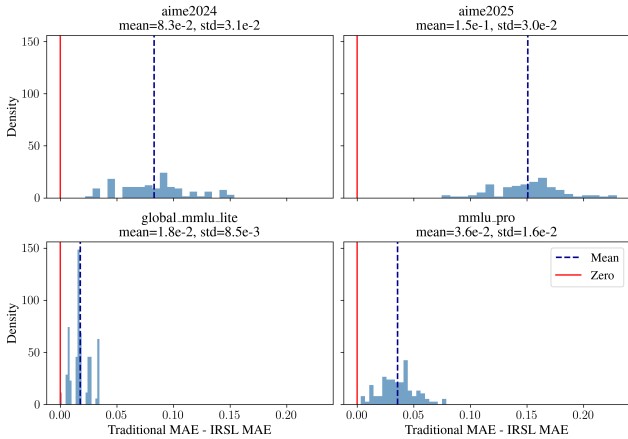

*Figure 8.* **IRSL consistently outperforms Traditional scaling across nearly all LM-benchmark pairs.** We visualize the distribution of the performance gap Traditional MAE $-$ IRSL MAE on four benchmarks across 100 random train-test splits. The distributions are consistently concentrated to the right of the zero line (red line), which indicates that IRSL achieves a lower MAE and thus provides a more accurate estimate.

per question. Given the limited number of LMs for calibration, we report the 1PL model as our primary result and present the 2PL findings in Appendix E[3].

We report three scaling curves ($-\log \text{pass@k}$ versus the number of samples $k$) for LMs in the test set in Figure 7: (1) The Ground Truth curve, where $\text{pass@k}$ is estimated from all available samples using the unbiased and numerically stable estimator proposed by Chen et al. (2021): $\text{pass@k}(i, j) \approx 1 - \binom{H - c_{ij}}{k} / \binom{H}{k}$, where $H$ is the total number of samples and $c_{ij}$ is the number of correct samples by LM $i$ on question $j$. (2) The traditional scaling curve, where $\text{pass@k}$ is estimated from the limited query budget via Equation 2. (3) The IRSL curve, where the

---

[3]The 2PL model typically requires more test takers to achieve reliable calibration.

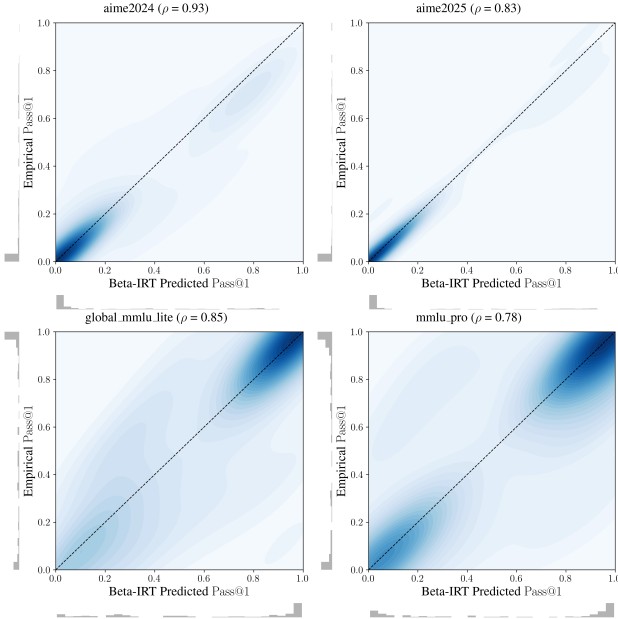

*Figure 9.* **Beta-IRT predicted** $\text{pass@1}$ **strongly correlates with empirical** $\text{pass@1}$ **across all test-time benchmarks.** Correlation between Beta-IRT 1PL predicted $\text{pass@1}$ (x-axis) and empirical $\text{pass@1}$ (y-axis), visualized using 2-D KDE contour plots. The Pearson correlation coefficient ($\rho$) is reported for each benchmark. The corresponding results for the 2PL variant are provided in Figure 24.

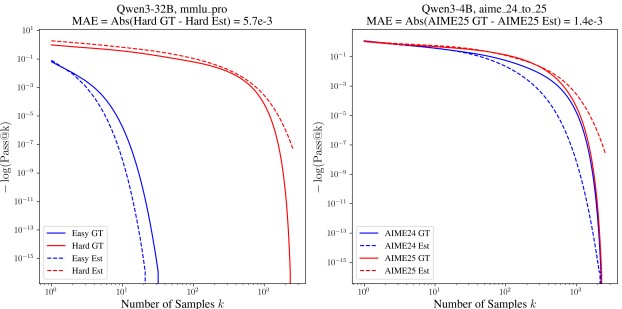

*Figure 10.* **Test-time IRSL ability transfers reliably from easy to hard sets and across benchmarks.** (Left) Within-benchmark transfer on MMLU Pro. (Right) Cross-benchmark transfer from AIME 2024 to AIME 2025. The close alignment between Hard GT and Hard Est demonstrates that the test-time scaling trend on harder sets can be reliably forecasted using ability parameters estimated from the easy sets.

ability $\theta$ is estimated from the same limited query budget, and $\text{pass@k}$ is subsequently derived using Equation 4. As shown in Figure 7, there is a high alignment between the IRSL curve and the Ground Truth curve. To quantify the superiority of IRSL against traditional scaling law, we compute the MAE of $-\log \text{pass@k}$ for both methods relative to the ground truth. We visualize the distribution of the performance gap Traditional MAE $-$ IRSL MAE in Figure 8 across all benchmarks and test LMs over 100 random train-

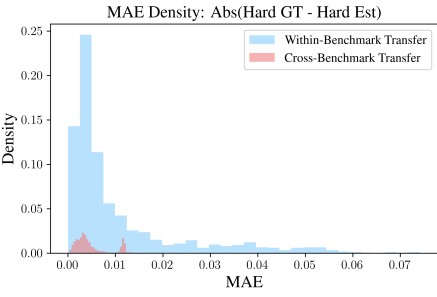

*Figure 11.* **Consistently low MAE confirms that test-time IRSL ability is transferable across difficulty levels.** We report the MAE between the ground truth scaling curve and the estimated curve for two settings: Within-Benchmark Transfer (blue) and Cross-Benchmark Transfer (red). The consistent low MAE values indicate that the ability $\theta$ estimated by IRSL enables reliable performance forecasting on benchmark sets with the same measurement objective.

test splits. The performance gap is predominantly positive, confirming that IRSL yields more reliable test-time scaling law estimates given a limited query budget.

We report the strong correlation between Beta-IRT predicted $\text{pass@1}$ and the empirical $\text{pass@1}$ on the test set, as illustrated in Figure 9 for the 1PL variant and Figure 24 for the 2PL variant. We further report the Beta-IRT curve on single questions in Figure 25 and 26 in Appendix E.

Next, we validate the generalizability of test-time IRSL across benchmark sets with different difficulty levels, following the same partitioning strategy used in our pre-training analysis. We estimate the ability $\theta$ using only the easy subset and transfer it to predict the scaling curve of the hard subset (or a harder benchmark) without accessing the response data. Figure 10 illustrates this capability: the left panel shows within-benchmark transfer for MMLU Pro using Qwen3-32B, while the right panel demonstrates cross-benchmark transfer, where $\theta$ estimated on AIME 2024 effectively predicts performance on AIME 2025. To quantify robustness across all settings, Figure 11 reports the distribution of the MAE between ground truth and estimated scaling curves for the hard sets across all benchmarks and LLMs over 100 random train-test splits. The consistently low errors confirm that the ability parameters $\theta$ estimated by IRSL are robustly transferable, enabling reliable test-time forecasting on harder tasks sharing the same measurement objective.

## 5. Limitations, Discussions, and Future Work

IRSL excels when benchmarks have heterogeneous question difficulty, evaluation budgets are limited, and cross-question generalization is needed. However, traditional scaling with $p_{\text{Correct Choice}}$ already performs well on high-

quality benchmarks with smooth probability responses (e.g., ARC Challenge, MMLU). In such cases, IRSL offers comparable accuracy with added interpretability but may not justify calibration overhead if only aggregate metrics are needed. On extremely noisy benchmarks (e.g., BoolQ, HellaSwag), neither approach captures reliable trends. Unlike classical power-law models that extrapolate to unseen compute regimes, IRSL requires pre-calibrated item difficulties from prior model responses, limiting applicability to established benchmarks. Difficulties calibrated under one evaluation setup may also not transfer to different conditions. IRSL is thus best viewed as complementary to traditional scaling laws. Besides, the restricted data scale for test-time scaling analysis is another primary limitation of this work.

In this work, we demonstrate that empirical probability information (either from noisy probability observations or from repeated sampling) provides additional signals that compensate for a limited number of test takers. In human testing, a test-taker sample size of 100 is typically insufficient for IRT, and practitioners can easily recruit more human test-takers. In contrast, LMs are relatively homogeneous (Kim et al., 2025) and limited in number due to query cost. However, LMs naturally provide token probabilities and support repeated sampling, which are not feasible in human testing. A key insight is that, to achieve robust estimation, human testing increases the number of test takers, whereas LM evaluation leverages empirical probability.

Future work includes scaling up the test-time experimental setup, fitting shared latent abilities across benchmarks (Truong et al., 2025; Kipnis et al., 2025), exploring alternative probabilistic models (e.g., Beta-Binomial, zero-inflated models), extending to other scaling laws (Ruan et al., 2024; Kaplan et al., 2020; Arora et al., 2025), and polytomous IRT (Ostini & Nering, 2006).

## Impact Statement

This paper presents work whose goal is to advance the field of Machine Learning. There are many potential societal consequences of our work, none which we feel must be specifically highlighted here.

## Acknowledge

SK acknowledges support by NSF 2046795 and 2205329, IES R305C240046, ARPA-H, the MacArthur Foundation, Schmidt Sciences, HAI, OpenAI, Microsoft, and Google.

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

## A. Pre-training Downstream Scaling Law Metrics Calculation Details

In this section, we explain the calculation of the benchmark-specific loss $L$, accuracy $\text{Acc}$, and the average probability of the correct choice $p_{\text{Correct Choice}}$. Consider a question from a multiple-choice benchmark:

> [Question Content]
> A. [Choice A Content]
> B. [Choice B Content]
> C. [Choice C Content]
> D. [Choice D Content]

Assuming the correct answer is C, the metrics are calculated as follows:

- **Benchmark-specific Loss:** Also known as bits per byte (BPB). For an individual question, this is calculated as the negative log-likelihood of the token sequence corresponding to the correct choice content (i.e., [Choice C Content]) conditioned on the question content (i.e., [Question Content]), normalized by the length of the correct choice content in bytes. The benchmarked-level value is averaged across all questions.

- **Average Probability of Correct Choice:** For an individual question, this measures the probability of the token sequence representing the correct choice content, conditioned on the question content, normalized by the character length of the choice. The benchmarked-level value is averaged across all questions.

- **Accuracy:** Also known as cloze formulation accuracy or RC format accuracy. This is determined by computing the probability of the token sequence for each choice content given the question content, normalized by the character length of each choice. The choice with the highest probability is selected as the predicted answer. The question is assigned a score of 1 if the prediction matches the correct choice, and 0 otherwise. The benchmarked-level value is averaged across all questions.

## B. Additional Results for Pre-training Downstream IRSL

Figure 12 shows the empirical observation of the linear relationship between $\theta$ and $\log \text{FLOP}$ for Beta-IRT 2PL. The trend is similar for Binary-IRT and 1PL variants.

Figure 13 shows the scaling curve fitting for traditional scaling law step 1. Figure 14 shows the scaling curve fitting for traditional scaling law step 2. Figure 15 shows the scaling curve fitting for IRSL step 1. Following Bhagia et al. (2024), we fit step 1 only on final checkpoints for each model size, as the learning rate schedule prevents accurate FLOP estimation on intermediate checkpoints.

Figure 16 shows the correlation between Beta-IRT 1PL predicted $p_{\text{Correct Choice}}$ and empirical $p_{\text{Correct Choice}}$. Figure 17 and 18 show the Beta-IRT curve on a randomly sampled question for 2PL and 1PL, respectively.

Figure 19 reports the MAE of hard set estimation across all benchmarks and LM data mixtures.

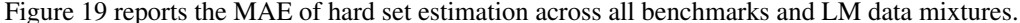

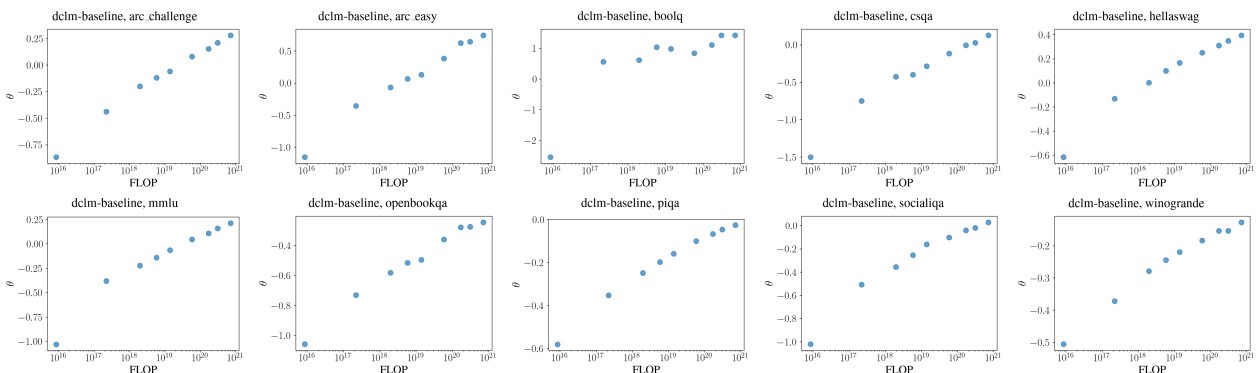

*Figure 12.* $\theta$ **scales linearly with** $\log \text{FLOP}$ **across all benchmarks.** Beta-IRT 2PL on the test set for a representative LM data mixture across all 10 benchmarks. This linear trend is consistent across other data mixtures, as well as Binary-IRT and 1PL variants.

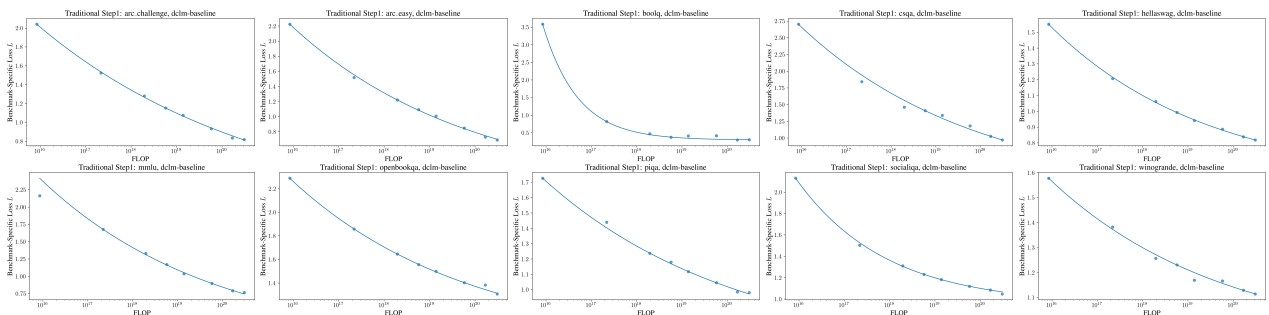

*Figure 13.* **Traditional scaling law step 1:** $L \approx \alpha \cdot \text{FLOP}^{-\beta} + \gamma$. Representative LM data mixture across all 10 benchmarks. The trend is consistent across other data mixtures.

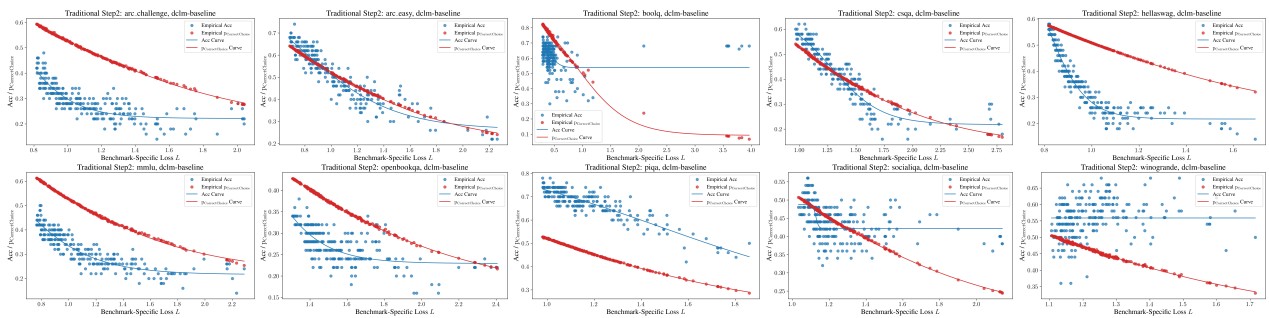

*Figure 14.* **Traditional scaling law step 2:** $\text{Performance}(i, \mathcal{D}) \approx a \cdot \sigma(b \cdot (L - l_0)) + c$. Representative LM data mixture across all 10 benchmarks. The trend is consistent across other data mixtures.

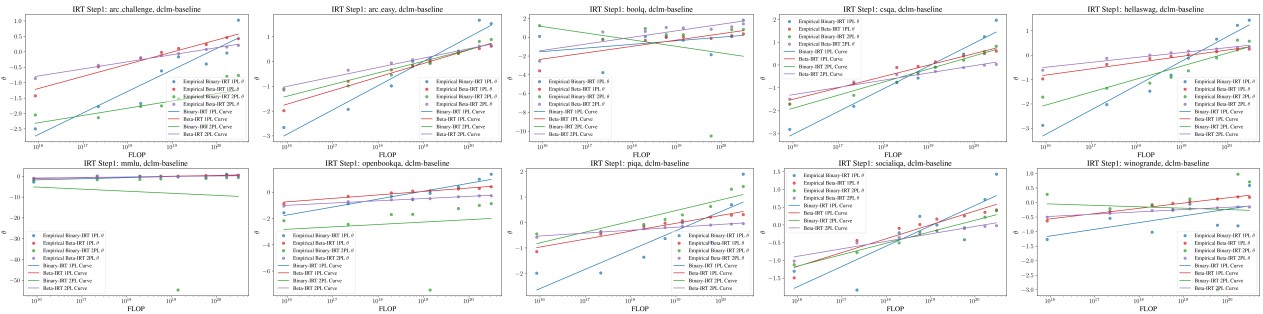

*Figure 15.* **IRSL step 1:** $\theta_i \approx a \cdot \log(\text{FLOP}_i) + b$. Representative LM data mixture across all 10 benchmarks. The trend is consistent across other data mixtures.

## C. Benchmark Homogeneity Inspection for Pre-training Downstream IRSL

To further explain why IRSL does not consistently outperform traditional scaling laws on certain benchmarks, we carry out an additional experiment on benchmark homogeneity. Figure 20 presents four rows of diagnostics per benchmark: (1) a response matrix heatmap of models versus questions, colored by probability of correct response; (2) item difficulty distribution; (3) item discrimination distribution; and (4) the Test Information Function (TIF): a measure of how precisely a benchmark estimates model ability at each point on the ability scale, computed as the sum of individual item information functions, where each item contributes more when its discrimination is high and its difficulty is well-matched to the ability being estimated. The shaded region marks the 5th–95th percentile of actual model abilities.

On BoolQ and HellaSwag, the response matrices (row 1) show almost no structural gradient, consistent with their narrow difficulty distributions (row 2; standard deviation of 0.19 and 0.29, respectively) and low discrimination spread (row 3; standard deviation of 0.14 and 0.30). Because items are nearly homogeneous in both difficulty and discrimination, the aggregate TIF (row 4) is low and flat.

ARC-Challenge presents a starkly different picture. The response matrix (row 1) shows clear diagonal stratification: a

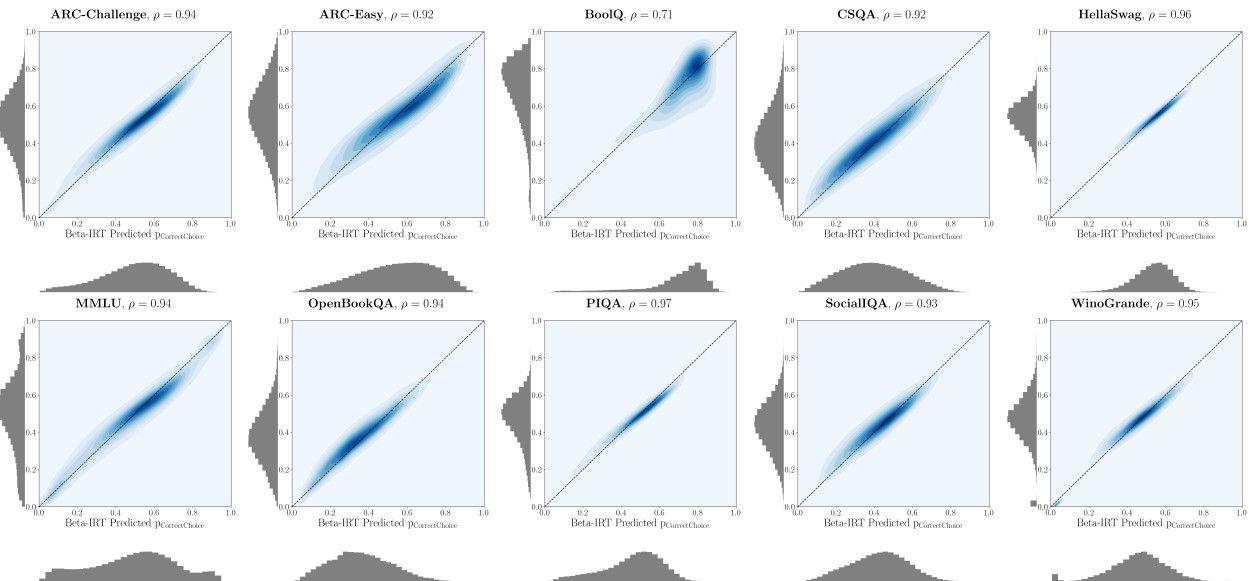

*Figure 16.* **Beta-IRT 1PL predicted** $p_{Correct\ Choice}$ **correlates strongly with empirical** $p_{Correct\ Choice}$.

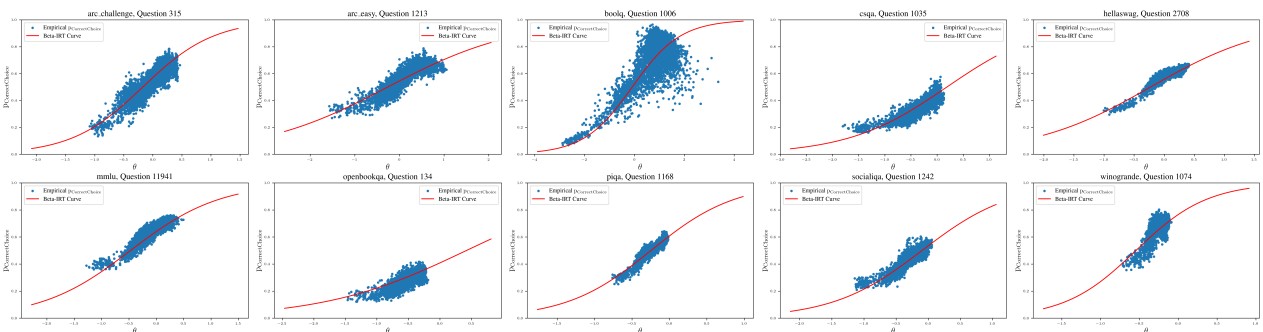

*Figure 17.* **Beta-IRT 2PL curve on a single question for each benchmark.** The x-axis is the ability parameter $\theta$, and the y-axis is $p_{Correct\ Choice}$. The red line shows the fitted Beta-IRT curve. The blue dots represent the empirical $p_{Correct\ Choice}$; each dot corresponds to an LM checkpoint in the test set.

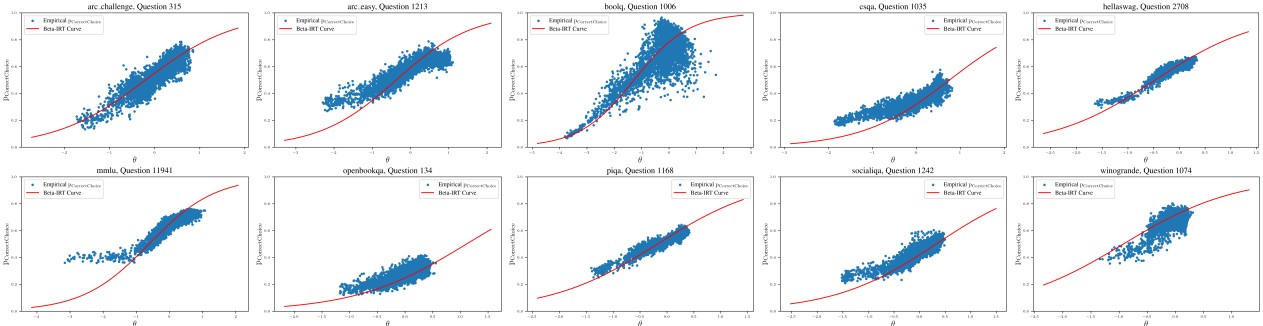

*Figure 18.* **Beta-IRT 1PL curve on a single question for each benchmark.**

smooth gradient from easy to hard items. Both the difficulty (standard deviation of 0.55) and discrimination (standard deviation of 0.61) distributions are substantially wider (rows 2–3). As a result, the TIF (row 4) exhibits a pronounced peak.

We therefore view this not as a limitation of IRSL, but as a property of the benchmarks themselves. IRSL is most effective when evaluation items are sufficiently diverse and informative, and we believe this finding itself contributes toward more

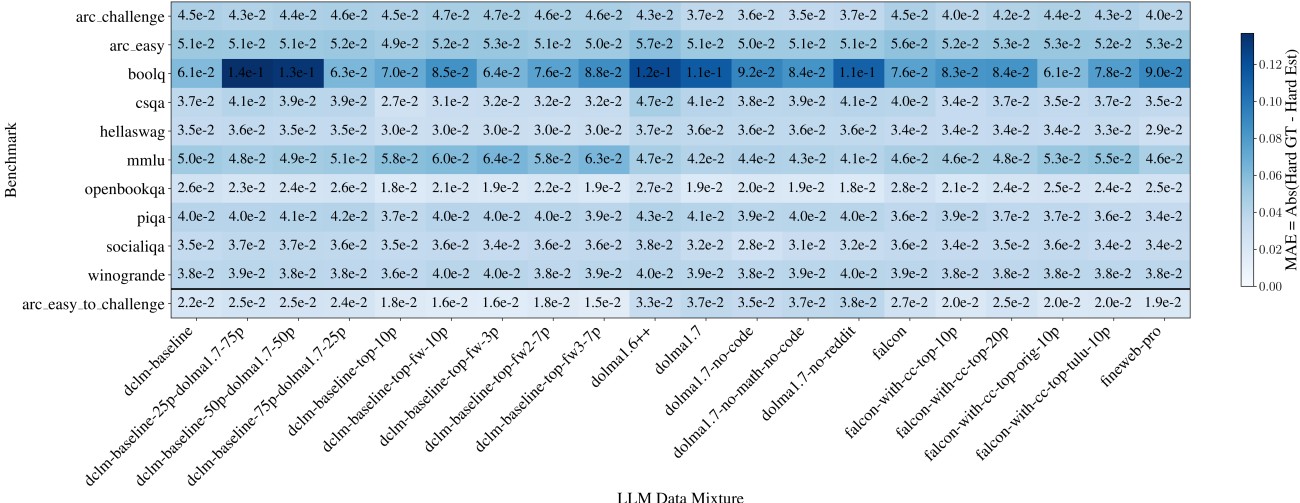

*Figure 19.* **MAE of hard set estimation across all benchmarks and LM data mixtures.** We report the MAE between the ground truth scaling curve and the estimated curve on the hard sets. The last row specifically corresponds to the cross-benchmark transfer from ARC Easy to ARC Challenge.

principled benchmark design.

## D. Construct Similarity and Cross-Benchmark Transfer

We mention that the LM ability $\theta$ estimated from one benchmark can transfer to another benchmark with similar measurement objectives. In this section, we show how to examine if two benchmarks share similar measurement objectives from two complementary perspectives: empirical convergent validity and benchmark design.

**Empirical convergent validity.** We provide direct empirical evidence via correlation plots of estimated LM ability $\theta$ across models. As shown in Figure 21, there are strong correlations between latent abilities estimated from paired benchmarks: $\rho = 0.99$ between ARC Easy and ARC Challenge (pre-training) and $\rho = 0.80$ between AIME 2024 and AIME 2025 (test-time). This confirms that these pairs share a consistently measured latent construct. We extend this analysis to all benchmark pairs in Figure 22. The full correlation heatmap shows that most of the 10 pre-training benchmarks exhibit high pairwise $\theta$ correlations, with BoolQ as the notable exception (BoolQ is known to have a low signal-to-noise ratio as a two-choice benchmark (Heineman et al., 2025)). This aligns with findings from Kipnis et al. (2025) that a single common factor underlies most benchmark scores, suggesting that cross-benchmark transfer may be more broadly applicable. For the test-time benchmarks shown in Figure 23, correlations are weaker, likely due to the limited experimental scale.

**Construct similarity as a design property.** Beyond empirical validation, we argue that construct similarity is often a design-level property established prior to evaluation. The relevant construct (e.g., mathematical reasoning, coding, domain-specific knowledge, or general capability) is defined upfront by whoever designs or uses the benchmark. ARC Easy and ARC Challenge were explicitly built to assess the same scientific reasoning construct at different difficulty levels; AIME 2024 and 2025 share identical format and objectives. This is analogous to how the community treats yearly administrations of standardized tests, such as the SAT, as measuring a consistent construct by design.

For arbitrary benchmark pairs without clear design-level similarity, we suggest that convergent validity should be empirically verified before transfer is attempted, for instance via the $\theta$ correlation analysis demonstrated above.

## E. Additional Results for Test-time IRSL

The 12 models used are: DeepSeek-R1-Distill-Llama-70B, DeepSeek-R1-Distill-Llama-8B, DeepSeek-R1-Distill-Qwen-14B, DeepSeek-R1-Distill-Qwen-32B, DeepSeek-R1-Distill-Qwen-7B, QwQ-32B, Qwen3-14B, Qwen3-30B-A3B, Qwen3-32B, Qwen3-4B, Qwen3-8B, and gemma-3-27b-it. The 4 benchmarks used are: AIME 2024, AIME 2025, Global MMLU Lite, and MMLU Pro.

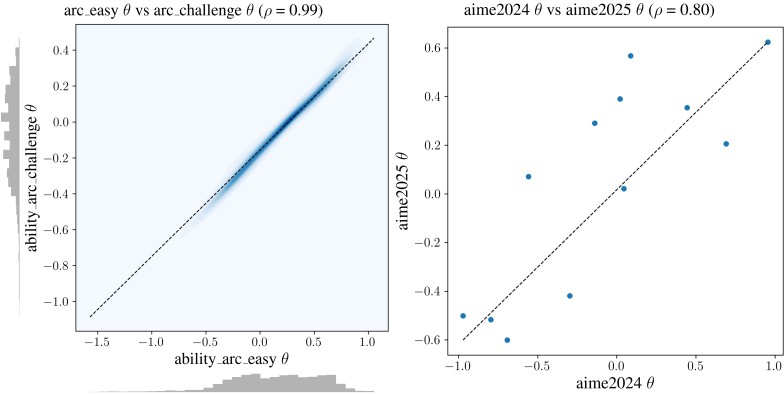

*Figure 20.* **Benchmark homogeneity analysis for BoolQ, HellaSwag, and ARC Challenge.** Top row: response matrix heatmaps with rows (models) sorted by mean $p_{\text{Correct Choice}}$ and columns (items) sorted by calibrated difficulty $z$. Middle rows: histograms of calibrated item difficulty $z$ and discrimination $d$. Bottom row: Test Information Function (TIF) per item, $I(\theta)/N$. BoolQ and HellaSwag exhibit highly concentrated item parameters ($\sigma_z$=0.19, 0.29; $\sigma_d$=0.14, 0.30), yielding near-uniform response matrices and low per-item information. In contrast, ARC Challenge shows diverse item parameters ($\sigma_z$=0.55, $\sigma_d$=0.61) and substantially higher TIF, enabling IRT to differentiate model abilities effectively.

*Figure 21.* **The transfer benchmark pairs exhibit strong convergent validity in estimated LM ability.** The x-axis shows the estimated ability $\theta$ on the source benchmark, and the y-axis shows the estimated ability $\theta$ on the target benchmark.

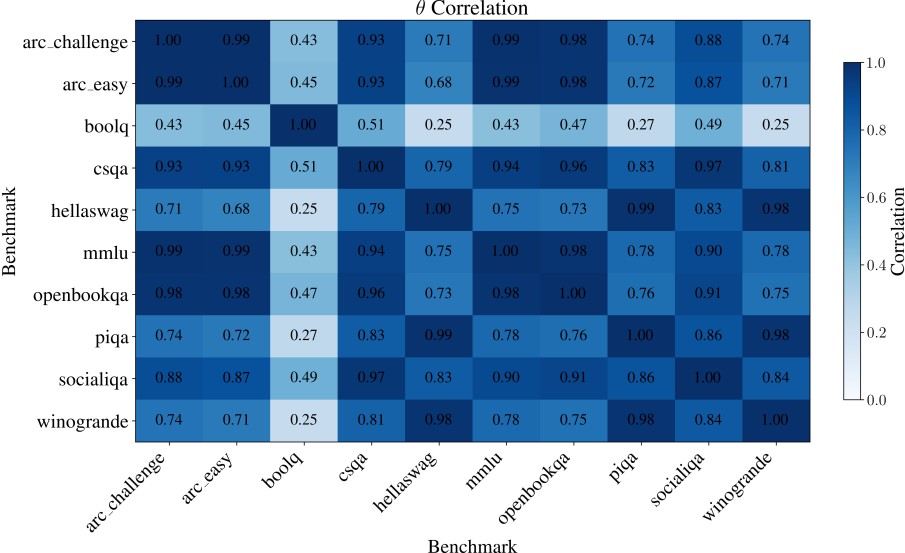

*Figure 22.* **Most pre-training benchmarks share a strongly aligned latent ability.** The x-axis and y-axis show pre-training benchmarks, and each cell reports the Pearson correlation of estimated ability $\theta$ between the corresponding benchmark pair.

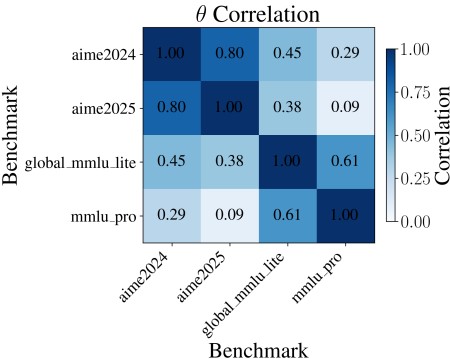

*Figure 23.* **Test-time benchmark abilities show weaker but still informative cross-benchmark alignment.**

Figure 24 shows the correlation between Beta-IRT 2PL predicted pass@1 and empirical pass@1. Figure 25 and Figure 24 show the Beta-IRT curve on a randomly sampled question for 1PL and 2PL, respectively.

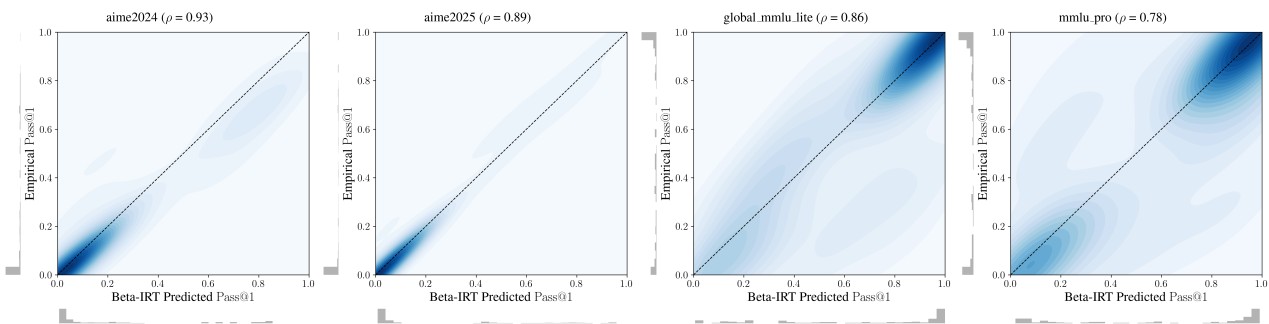

*Figure 24.* **Beta-IRT 2PL predicted** pass@1 **correlates strongly with empirical** pass@1**.**

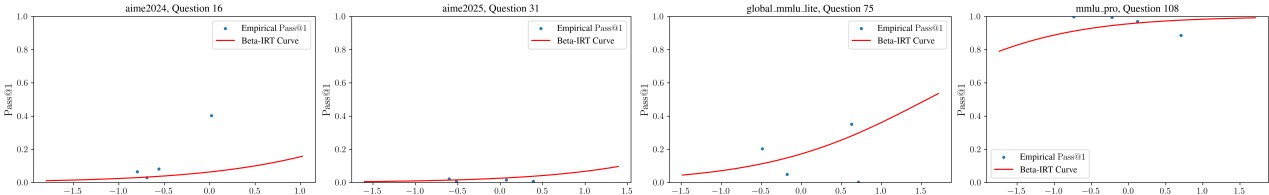

*Figure 25.* **Beta-IRT 1PL curve on a single question for each test-time benchmark.** The x-axis is the ability parameter $\theta$, and the y-axis is pass@1. The red line shows the fitted Beta-IRT curve. The blue dots represent the empirical pass@1; each dot corresponds to an LM in the test set.

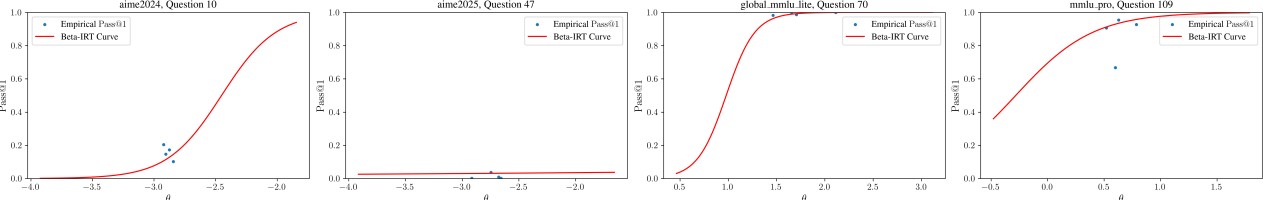

*Figure 26.* **Beta-IRT 2PL curve on a single question for each test-time benchmark.**

