# OpenReview forum: "Item Response Scaling Laws: A Measurement Theory Approach for Efficient and Generalizable Neural Scaling Estimation"
_ICML.cc/2026/Conference — ICML 2026 regular_

### Official Review · Reviewer_adSR · 2026-02-19

**Soundness:** 3
**Presentation:** 4
**Significance:** 3
**Originality:** 4
**Overall Recommendation:** 4
**Confidence:** 4

**Summary:**

This paper incorporates Item Response Theory (IRT) from psychometrics to improve the parameter and computational efficiency of scaling law fitting.
Specifically, the authors first replace the standard Bernoulli loss used in typical IRT calibration with a Beta loss to model the empirical probability responses from LLMs, then propose Item Response Scaling Laws (IRSL) on top of the Beta-IRT to disentangle model abilities and benchmark properties in scaling law fitting.
A simulation study suggests that IRSL reduces the complexity from O(MN) to O(M+N) without compromising the accuracy and generalizability. Experiments also validate IRSL's advantage in pre-training downstream scaling and test-time scaling (at a relatively smaller scale).

**Compliance With Llm Reviewing Policy:**

Affirmed.

**Final Justification:**

I recommend this paper to be accepted.

**Key Questions For Authors:**

1. For the empirical probability responses, one of my questions is that those responses are inherently probabilistic. Therefore, if the traditional IRT is applied in the context of LLM evaluation, usually the responses should be binarized to fit into the framework. Then, I am not sure if it is more appropriate to attribute the sample efficiency of Beta-IRT to its compatibility with continuous responses or to other algorithmic design.
2. From my understanding, with only one sample, pass@1 can only be 0 or 1, as there are only two outcomes (*pass* and *fail*), and more samples are needed for a more accurate estimate of pass@1 and pass@k. What if we compare a traditional IRT model calibrated on sample-level binary responses with a Beta-IRT model calibrated on aggregated pass@k results?
3. Given the nature of IRT, it is reasonable that only the model parameters can be transferred across similar benchmarks. How can we know whether two benchmarks are similar in objectives or constructs? Does this require measuring the convergent validity between benchmarks?
4. Could the authors provide more insights on how this interdisciplinary framework could be extended and applied to real-world, practical cases?

**Limitations:**

Yes.

**Strengths And Weaknesses:**

### **Strengths**
1. The paper adopts an interdisciplinary perspective and meaningfully incorporates two theoretical frameworks from psychometrics and AI; both IRT and scaling laws are properly adapted to be mutually compatible.
2. Beta-IRT is shown to leverage the fine-grained signals in LLM responses, which makes it more sample-efficient than traditional binary-IRT in the context of LLM evaluation.
3. A systematic evaluation of IRSL is conducted to demonstrate the effectiveness and transferability of pre-training downstream IRSL with limited budget.
4. The theoretical derivations are rigorous, and the experimental results are comprehensive and straightforward.

### **Weaknesses**
1. In Fig. 3, IRSL doesn't appear to outperform traditional scaling laws on *boolq*, *hellaswag*, and *MMLU*. Although the study of the signal-to-noise ratio may partly account for this in that these benchmarks might be noisy, they are more general and closer to in-the-wild data than other benchmarks. The authors are encouraged to provide further explanations for this phenomenon.
2. In Sec. 3.1, with the same number of responses per test-taker per item, the informativeness of Y_ij would be inherently less than that of P_ij, which is a significant confounder in the comparison. There should be some practices to minimize the impact of this confounder, for example, adding a setting where the binary response matrix is aggregated into a smaller P_ij.

---

> ### Author Rebuttal · Authors · 2026-03-31
>
> **Answer to Weakness 1:** We agree further explanation is needed for why IRSL does not consistently outperform traditional scaling on some benchmarks, which we attribute to benchmark homogeneity. The [Figure](<https://anonymous.4open.science/r/irsl_rebuttal_figures-4205/boolq_hellaswag_vs_arc.png>) shows four diagnostics per benchmark: (1) response matrix heatmap; (2) difficulty distribution; (3) discrimination distribution; and (4) the Test Information Function (TIF), which measures ability estimation precision across the ability scale as the sum of item information, higher when items have high discrimination and well-matched difficulty (shaded region: 5th–95th percentile of actual abilities).
>
> On BoolQ and HellaSwag, response matrices show almost no structural gradient, with narrow difficulty and discrimination spread, yielding a flat TIF. In contrast, ARC-Challenge shows clear stratification, wider distributions, and a peaked TIF.
>
> We thus view this as a benchmark property, not an IRSL limitation: IRSL is most effective when items are diverse and informative. We believe this finding informs more principled benchmark design, and we will add this discussion to the revision.
>
> **Answer to Weakness 2:** Thank you for the question. Due to limited space, we refer to our response to weakness 3 for reviewer 7jJA.
>
> **Answer to Question 1:** We agree that the improvement stems from the richer information in continuous responses, and we view compatibility with continuous probabilities as a key feature of Beta-IRT. Unlike human test-takers, LMs do not produce inherently binary responses; binarization is an artificial step that discards information. Binary-IRT thus imposes an unnecessary constraint, which Beta-IRT removes via a Beta-loss framework suited for responses in (0,1).
>
> **Answer to Question 2:** Your understanding is correct, and your suggested comparison is well-motivated. We conducted an additional experiment reporting the distribution of (Binary-IRT MAE − Beta-IRT MAE) over 100 random train-test splits, where MAE measures error to the ground truth scaling curve. As shown in [Figure](<https://anonymous.4open.science/r/irsl_rebuttal_figures-4205/beta_vs_binary_irt_testtime.png>), the difference is consistently positive, confirming Beta-IRT’s superior performance. We will add this to the revision.
>
> **Answer to Question 3:** Thank you for the question. We address this from two angles.
>
> 1. **Empirical evidence.** We analyze correlations of estimated LM ability $\theta$ across benchmarks. As shown in [Figure Pretrain](<https://anonymous.4open.science/r/irsl_rebuttal_figures-4205/pretrain_thetacorr_arc.jpeg>) and [Figure Testtime](<https://anonymous.4open.science/r/irsl_rebuttal_figures-4205/testtime_thetacorr_aime.png>), paired benchmarks exhibit strong $\theta$ correlations, indicating a shared latent construct. Extending to all pairs ([Figure](<https://anonymous.4open.science/r/irsl_rebuttal_figures-4205/pretrain_thetacorr_all.jpeg>)), most of the 10 pre-training benchmarks show high correlations, with BoolQ as an exception (low signal-to-noise two-choice benchmark according to Heineman et al. (2025)). This aligns with Kipnis et al. (2025), suggesting a single factor underlies most benchmark scores. Thus, cross-benchmark transfer may be more broadly applicable than our original submission presents. In the test-time scenario ([Figure](<https://anonymous.4open.science/r/irsl_rebuttal_figures-4205/testtime_thetacorr_all.png>)), correlations are weaker, likely due to limited scale.
>
> 2. **Design perspective**. Construct similarity is often defined a priori by benchmark creators and users (e.g., ARC Easy/Challenge and AIME 2024/2025 measure the same construct at different difficulty levels), analogous to standardized tests like the SAT.
>
> For arbitrary benchmark pairs, we agree convergent validity should be verified empirically (e.g., via $\theta$ correlation) before transfer is attempted. We will clarify this in the revision.
>
> Heineman et al. Signal and noise: A framework for reducing uncertainty in language model evaluation. NeurIPS 25
>
> Kipnis et al. metabench--A Sparse Benchmark of Reasoning and Knowledge in Large Language Models. ICLR 25.
>
> **Answer to Question 4:** Thank you for your question on real-world use cases.
>
> Pre-training data selection is a key application. Our downstream IRSL experiments (Sec. 3.2) directly target a high-stakes real-world decision: which data mixture to scale for full pre-training; by reliably and efficiently estimating scaling laws, IRSL enables better-informed choices.
>
> For test-time scaling, a concrete use case is model selection under repeated querying. On tasks like code generation with unit-test verifiers, many samples are needed for stable pass@k estimates. IRSL reduces the otherwise prohibitive models × questions × samples evaluation cost.
>
> We will add these practical implications to the revision.

---

> > ### Author Rebuttal · Reviewer_adSR · 2026-04-02
> >
> > Thanks for the response!

---

> > > ### Author Response · Authors · 2026-04-06
> > >
> > > Thank you very much for your time and constructive suggestion. We sincerely appreciate your support!

---

### Official Review · Reviewer_Btax · 2026-03-07

**Soundness:** 3
**Presentation:** 3
**Significance:** 2
**Originality:** 3
**Overall Recommendation:** 4
**Confidence:** 3

**Summary:**

This paper proposes Item Response Scaling Laws, integrateing Item Response Theory into neural scaling law estimation. The main idea is to decompose benchmark performance into latent model ability and item-level question characteristics. The paper studies both pre-training and test-time scaling, and shows that the proposed framework can provide more reliable scaling estimates under limited query budgets.

**Compliance With Llm Reviewing Policy:**

Affirmed.

**Final Justification:**

My concerns have been adequately addressed.

**Key Questions For Authors:**

Please refer to Weakness

**Limitations:**

Please refer to Weakness

**Strengths And Weaknesses:**

Strength:
- the authors investigate the concept of reframing scaling law estimation through IRT rather than fitting only aggregate benchmark-level curves, which is interesting
-  The large-scale pre-training study is also a good

Weekness:

-  The proposed framework relies on calibrated item parameters, which makes its practical generalization less clear in truly new benchmarks or settings without prior calibration. If my understanding is correct, this means the method is more naturally viewed as an efficient measurement framework rather than a fully general predictive scaling-law framework. As a result, its practical usefulness may be more limited than the paper currently suggests. This is my main concern. To strengthen the claim, the paper would benefit from more explicit discussion and comparison against other efficient and generalizable model evaluation approaches, so to make IRSL provides a distinct or more solid advantage.

- Relatedly, the practical meaning of "generalizable" is somewhat limited in the current paper.

---

> ### Author Rebuttal · Authors · 2026-03-31
>
> Dear Reviewer Btax,
>
> Thank you for your valuable feedback. We answer your comments below.
>
> **Answer to Weakness 1:** Thank you for raising the concern on the limitation of IRSL requiring calibration and the comparison with other methods. As we acknowledge in Section 4, requiring pre-calibrated item parameters is a limitation of our work. However, we would like to note that in practice, benchmarks are typically released alongside evaluations on multiple LMs, which can naturally serve as calibration data, potentially making the requirement less restrictive than it may appear. Moreover, even for benchmarks where such evaluations are unavailable, calibration only requires a small set of models, meaning the initialization cost is trivial compared to the scaling phase. Crucially, because Beta-IRT leverages empirical probability information rather than binary responses, it requires far fewer LMs for reliable calibration than traditional IRT, further reducing this overhead.
>
> Regarding comparison with other efficient evaluation approaches: Binary-IRT has been extensively validated in the LLM evaluation setting (Polo et al., 2024; Hofmann et al., 2025), where it has been shown to outperform many efficient evaluation methods, such as Anchor Points (Vivek et al., 2024), SMART (Gupta et al., 2024), MAGI (Paech, 2024), and Stratified Sampling (Perlitz et al., 2024). Our contribution integrates this framework into the scaling law estimation scenario and further uses Beta-IRT, which leverages empirical probability responses unique to LMs to achieve better performance than Binary-IRT. We will add this to the Related Work section of the revised paper.
>
> Hofmann, Valentin, et al. "Fluid language model benchmarking." COLM 25.
>
> Polo, Felipe Maia, et al. "tinyBenchmarks: evaluating LLMs with fewer examples." ICML 24.
>
> Vivek, Rajan, et al. "Anchor points: Benchmarking models with much fewer examples." ACL 24.
>
> Gupta, Vipul, et al. "Improving model evaluation using smart filtering of benchmark datasets." ACL 25.
>
> Sam Paech. "Creating MAGI: A hard subset of MMLU and AGIEval." https://sampaech.substack.com/p/creating-magi-a-hard-subset-of-mmlu, 24.
>
> Perlitz, Yotam, et al. "Efficient benchmarking (of language models)." ACL 24.
>
> **Answer to Weakness 2:** Thank you for the insightful feedback regarding the practical interpretation of "generalizability." We acknowledge that the generalizability demonstrated in our paper is more of a scientific claim, showing that a shared latent ability underlies performance across benchmarks with the same measurement objective. However, we argue this has concrete practical value, particularly in the context of modern AI development where models and evaluations co-evolve rapidly. As models grow stronger, benchmarks saturate and are replaced by harder variants (e.g., MMLU → MMLU Pro, AIME 2024 → AIME 2025), meaning practitioners must continuously fit new scaling laws from scratch each time a new benchmark is introduced. Our framework addresses this directly: rather than treating each new benchmark as an entirely independent evaluation, IRSL treats them as test sets of varying difficulty that measure the same underlying model ability, and gives an efficient estimate of the scaling curve by transferring ability estimates between benchmarks. Furthermore, as discussed in our answer to question 3 for reviewer adSR, evidence suggests that a single common factor underlies most benchmark scores, implying that cross-benchmark transfer may be more broadly applicable than it appears in the current submission. We will clarify these practical implications more explicitly in the revised paper.

---

> > ### Author Rebuttal · Reviewer_Btax · 2026-04-02
> >
> > Thanks for the response. I'll keep my score, for it is high enough.
> >
> > Also, another efficient evaluation approach might be related:
> > EffiEval: Efficient and Generalizable Model Evaluation via Capability Coverage Maximization.

---

> > > ### Author Response · Authors · 2026-04-06
> > >
> > > Thank you very much for your time and constructive suggestion. We will cite and discuss EffiEval in the revised version.

---

### Official Review · Reviewer_7jJA · 2026-03-13

**Soundness:** 3
**Presentation:** 3
**Significance:** 3
**Originality:** 3
**Overall Recommendation:** 5
**Confidence:** 4

**Summary:**

This work combines two different paradigms for predicting LLM performance: IRT modeling and scaling laws. The core idea is to first fit IRT models onto the underlying data, and then learn scaling laws over the parameters of these models. This has the benefit of not only providing a more well motivated model of benchmark performance, but also enhanced sample efficiency and data fit. The authors fit their models using an extension of existing models Beta-IRT, which operates on success rates per item rather than a simple binary outcome. They analyze this approach for both pre-training scaling and test-time scaling, demonstrating superior performance to the standard baselines.

**Compliance With Llm Reviewing Policy:**

Affirmed.

**Final Justification:**

The authors rebuttal helped clarify some of the main concerns I had (particularly surrounding Beta-IRT). I think that overall the core idea and presentation of the paper warrants admission.

**Key Questions For Authors:**

1. Do you have a hypothesis as to why IRSL seems to perform relatively stronger on test-time rather than pre-training scaling compare to the baselines?
1. What were the train LLMs used in the scaling analysis? Multiple test models seem to be from the same family, were these represented in the test set as well?
1. Do you have a hypothesis on why Beta-IRT would help with lower quality datasets?

**Limitations:**

yes

**Strengths And Weaknesses:**

# Strengths
- The core idea is a clean connection between two disparate ways of thinking about how to predict LLM performance. Figure 12 is a wonderful visualization and clear demonstration of the motivation.
- Measuring Decision Accuracy is a clean experimental setup with a practical interpretation: "how well can I select the best data distribution?"
- The results for test-time scaling seem to demonstrate that the IRSL method consistently is able to outperform the traditional baselines.

# Weaknesses
- Related works should be in the main body of the paper.
- An IRT model with a continuous response or a beta likelihood is not entirely novel (e.g. "A Beta Item Response Model for Continuous Bounded Responses." Noel et al., 2007).
- Section 3.1 is not a fair comparison between the beta and binary methods. The empirical probability matrix naturally contains more information than the binary counterpart. However, binary IRT models can be fit on multiple responses for the same item. A more fair comparison would be to simulate sampling K examples per item and compare fitting on all K binary responses versus the single empirical probability estimated over those K samples.
- It is unclear what the error bars should be in Figure 3. These results do not seem to suggest that IRSL is consistently providing a better fit than the simpler, traditional methods. While it performs better in some cases, it is hard to disentangle true differences from noise.
- Test time scaling data relies on only 12 LLMs and a single train-test split.

---

> ### Author Rebuttal · Authors · 2026-03-31
>
> **Answer to Weakness 1:** Thank you for this suggestion. In the revised version, we will move the Related Work section from Appendix into the main body.
>
> **Answer to Weakness 2:** Thank you for these foundational references; we will incorporate them into our revised version. We want to clarify that our core novelty lies in the integration of Beta-IRT into the LM scaling law estimation framework and the empirical demonstration.
>
> **Answer to Weakness 3:**  Thank you for the question. We conduct an additional experiment: we generate a binary response tensor of shape (n_test-takers, n_questions, n_samples) from ground-truth IRT probabilities, then average over samples to obtain empirical probabilities. For Binary-IRT, we compute the Bernoulli loss over all n_samples observations for each (test-taker, question) pair. As shown in [Figure](<https://anonymous.4open.science/r/irsl_rebuttal_figures-4205/fair_synthetic.png>), in this setting, Binary-IRT performs comparably to Beta-IRT. Mathematically, sampling introduces binomial noise; as n_samples $\to \infty$, Binary-IRT converges to the true probability.
>
> We also want to clarify the goal of the original synthetic experiment: to show that empirical probability (from noisy observations or repeated sampling) provides additional signals to compensate for limited test-takers. The key insight is that, to achieve robust estimation, human testing increases the number of test takers, whereas LM evaluation leverages empirical probability:
> - In human testing, a test-taker size of ~100 is typically insufficient for IRT, but additional participants can be easily recruited.
> - In LM evaluation, the typical test-taker size is ~100 due to query costs, and models are homogeneous (Kim et al., 2025). However, LMs provide token probabilities and enable repeated sampling—signals unavailable in human testing. Prior work shows token probabilities and pass@k are highly correlated (Figure 6, Schaeffer et al., 2026), and token probabilities are cheaper to obtain than repeated sampling.
>
> While both Binary- and Beta-IRT can leverage repeated sampling, Beta-IRT further leverages cheaper token probabilities. In practice, one can use Beta-IRT on token probabilities or pass@1, or Binary-IRT on sampled binary responses. We will clarify this in the revision.
>
> Kim et al. Correlated errors in large language models. ICML 25
>
> Schaeffer et al. Pretraining Scaling Laws for Generative Evaluations of Language Models. ICLR 26
>
> **Answer to Weakness 4:** We agree that distinguishing true differences from noise is important. We update Figure 3 with error bars showing std over 5 train–test splits at [Figure](<https://anonymous.4open.science/r/irsl_rebuttal_figures-4205/decision_acc_5seed.png>). The overall trends remain consistent, indicating the differences are not driven by noise.
>
> **Answer to Weakness 5:** Thank you for raising this concern. As noted in Section 4, the limited scale of test-time experiments is a limitation of our work, mainly due to compute constraints. Accordingly, we deliberately downweight our claims (e.g., in the introduction, we state: preliminary evidence suggests that IRSL similarly applies to test-time scaling).
>
> Regarding the single train-test split, we repeat the experiment over 100 random splits and update Figure 8 to show the distribution of (Traditional MAE − IRSL MAE). As shown in [Figure](<https://anonymous.4open.science/r/irsl_rebuttal_figures-4205/testtime_100seed.png>), the distribution is consistently right of zero (red line), indicating IRSL achieves lower MAE across all splits.
>
> **Answer to Question 1:** This is a perceptive observation. While direct comparison is nuanced due to different metrics (Decision Accuracy vs. MAE of −log pass@k), we hypothesize the stronger performance comes from task properties: test-time scaling tasks (e.g., math, code) have more diverse item parameters than pre-training benchmarks. As noted in our response to weakness 1 for reviewer adSR, IRSL gains are largest on such diverse, discriminative benchmarks.
>
> **Answer to Question 2:** We appreciate the question. In test-time setting, we use 12 models ([List](<https://anonymous.4open.science/r/irsl_rebuttal_figures-4205/testtime_lms.png>)); we will include the table in Appendix.
>
> Regarding family overlap in the test split: the original submission uses a single random split, which may include same-family models. To address this, we repeat experiments over 100 random splits (see answer to weakness 5) and observe consistent results.
>
> **Answer to Question 3:** We hypothesize Beta-IRT is effective on lower-quality datasets due to a “soft label” smoothing effect. Such datasets contain ambiguous or noisy questions, where Binary-IRT assigns hard 0s that overly penalize likelihood and distort ability estimates. By modeling continuous probabilities, Beta-IRT avoids aggressive updates to discrete failures, yielding more robust scaling fits under high item-level variance.

---

> > ### Author Rebuttal · Reviewer_7jJA · 2026-04-03
> >
> > W2: I agree that the paper has novelty further than the Beta-IRT model. The current wording in the abstract does not align with that, however. "We propose Beta-IRT, a novel extension..." I am not entirely convinced that this work requires Beta-IRT.
> >
> > W3: I appreciate running the additional experiment, those results align with my expectations. Thank you for that clarification, that framing makes more sense to me.
> >
> > W5: This convinces me that its not sensitive to the split. I would still encourage a greater diversity of underlying models. It seems like most are Qwen/Qwen-derivatives? This lack of diversity still lowers the validity of the experiment.
> >
> > Overall, this response did aid in the areas I was uncertain. I will increase my score to Accept.

---

> > > ### Author Response · Authors · 2026-04-06
> > >
> > > Thank you for increasing your score and for your continued helpful advice.
> > >
> > > W2: We agree with your insight regarding the abstract's framing. Our primary contribution is indeed the unified Item Response Scaling Law (IRSL) framework, and our current wording slightly obscures that focus. To better align the abstract with our core novelty, we will update the text in our revision from "We propose Beta-IRT..." to "We leverage Beta-IRT...". Regarding whether this work requires Beta-IRT: while IRSL could theoretically operate on standard Binary-IRT in the test-time scenario, leveraging continuous empirical probabilities via Beta-IRT is what practically unlocks the sample efficiency, especially for the pre-training scenario. We will make this clear in the revised version.
> > >
> > > W3: We are glad that the additional experiment addressed your concerns.
> > >
> > > W5: Thank you for your suggestion. We will clearly state the limited diversity of test-time models as a limitation of our work, and highlight increasing experiment scale and model diversity as important future directions.
> > >
> > > Thank you again for your constructive feedback, which has helped us significantly strengthen the paper!

---

### Official Review · Reviewer_XXm6 · 2026-03-13

**Soundness:** 2
**Presentation:** 3
**Significance:** 3
**Originality:** 3
**Overall Recommendation:** 4
**Confidence:** 2

**Summary:**

This paper addresses the significant computational overhead associated with evaluating Large Language Models (LLMs) during pre-training. It investigates how to derive accurate scaling curves under a strictly limited evaluation budget.

To solve this, the authors propose leveraging empirical probability responses, such as token-level probabilities. In contrast to traditional Item Response Theory (IRT) approaches that rely on binary (Boolean 0/1) outcomes , the introduced Beta-IRT framework utilizes a Beta loss function to model the continuous probability of the models' responses.

The proposed methodology is executed in two phases.

First, a small number of model checkpoints are utilized as anchors to evaluate all questions within a benchmark; this calibration phase determines the item parameters (such as question difficulty and discrimination).

Subsequently, for each remaining checkpoint, the framework requires evaluating only a limited subset of questions. By utilizing an adaptive testing approach, the system dynamically selects the most informative questions based on the current estimation of the model's latent ability, thereby efficiently approximating its overall capability with minimal queries.

**Compliance With Llm Reviewing Policy:**

Affirmed.

**Final Justification:**

Thanks to the author's rebuttal, I will keep my rating

**Key Questions For Authors:**

see weekness

**Limitations:**

yes

**Strengths And Weaknesses:**

Strengths:

* Innovative and Highly Practical Methodology: The integration of IRT into the formulation of test-time scaling laws is highly innovative. This approach elegantly addresses a critical bottleneck in LLM development by significantly optimizing evaluation resources. Crucially, it enables highly targeted, adaptive assessments that successfully reduce computational overhead without degrading the accuracy or reliability of the resulting scaling predictions.

* Exceptionally Solid Empirical Validation: The paper is backed by extensive and rigorous empirical work. The experiments—covering 6,612 LLM checkpoints and 37,682 questions for pre-training scaling, alongside test-time scaling evaluations—thoroughly validate the authors' theoretical claims. Furthermore, the demonstration of cross-benchmark generalization (transferring ability estimates from easy to hard sets) robustly confirms the framework's effectiveness.

Weaknesses

* Insufficient Justification of Practical Significance (Evaluation vs. Training Compute): While the paper heavily emphasizes the massive resource consumption required to evaluate scaling laws, it fails to provide a concrete estimation or comparison between the compute utilized for these evaluations versus the compute required for the actual model pre-training. There is a prevailing consensus in the community that training compute dwarfs evaluation compute by several orders of magnitude. Without a basic quantitative cost analysis to challenge this inherent perception, it is difficult to convince the reader that optimizing evaluation resources is a pressing bottleneck that warrants this complex intervention.

* Incomplete Discussion on Existing Continuous Evaluation Metrics: The authors build their core premise on the claim that current evaluations predominantly rely on discrete (0/1) binary responses. However, this overlooks standard practices where evaluations are already inherently continuous. For instance, metrics like PPL or the probability of correct choices in In-Context Learning (ICL) setups (which is exactly how benchmarks like MMLU are frequently scored) are widely used continuous indicators. Although the paper empirically compares Beta-IRT with a "Traditional $P_{Correct Choice}$" baseline, it lacks a sufficiently deep, principled discussion on the theoretical distinctions and specific advantages of Beta-IRT over these established continuous measurement paradigms.

* Structural Organization: The structure of the manuscript is somewhat unconventional and impedes readability. Placing the Method section immediately as Section 2 , while relegating the Background and Related Work to Appendix A, steepens the learning curve for readers unfamiliar with psychometric theories.

* Sample Complexity vs. Parameter Complexity: The authors claim a theoretical reduction in parameter complexity from $O(M \times N)$ to $O(M + N)$. However, this efficiency relies heavily on the adaptive testing stage utilizing a constant budget of queries (e.g., 50 questions or 50 samples) to estimate the latent model ability. It remains insufficiently discussed whether a constant sample size is statistically robust enough to accurately estimate this ability across benchmarks of varying sizes (a very large $N$) or benchmarks with high internal variance.

---

> ### Author Rebuttal · Authors · 2026-03-31
>
> Dear Reviewer XXm6,
>
> Thank you for your valuable feedback. We answer your comments below.
>
> **Answer to Weakness 1:** We appreciate the opportunity to clarify this critical motivation. We agree that pre-training compute generally dwarfs evaluation compute. However, evaluation is overwhelmingly the inner loop of modern LLM development. During pre-training, researchers evaluate hundreds of intermediate checkpoints across dozens of data mixtures (e.g., architecture ablations, hyperparameter sweeps) to decide what configurations to scale.
>
> Furthermore, as test-time compute scaling (e.g., reasoning models) becomes a primary frontier, evaluating scaling properties requires repeated sampling (often hundreds or thousands of times per item to estimate pass@k). This transforms inference-time evaluation into a massive computational bottleneck. Optimizing this inner loop via IRSL directly translates to dramatically faster iteration cycles and significant aggregate compute savings.
>
> **Answer to Weakness 2:** This is an important theoretical distinction that we will clarify in the revision. While standard metrics like raw Perplexity (PPL) or In-Context Learning (ICL) choice probabilities are indeed continuous, they inherently entangle the model's underlying capability with the specific idiosyncrasies and difficulty of the question. For example, a highly capable model might output a low probability simply because the prompt is ambiguous, not because the model lacks capability.
>
> Beta-IRT explicitly disentangles these factors into a latent model ability ($\theta$) and question characteristics (difficulty $z$, discrimination $d$). By isolating pure "model ability," IRSL provides a much cleaner, monotonic signal for fitting scaling laws. Predicting scaling curves on raw probabilities requires modeling complex, question-specific noise, whereas predicting scaling curves on $\theta$ models the true architectural capability.
>
> **Answer to Weakness 3:** We agree with this structural feedback. In the revised manuscript, we will move the Background and Related Work on IRT from Appendix A directly to Section 2. This will ensure readers have the requisite psychometric context before encountering the methodology, smoothing the learning curve.
>
> **Answer to Weakness 4:** Thank you for your insightful question on constant budget size. The estimation of latent ability using a constant budget (e.g., 50 questions) is statistically robust precisely because of the adaptive testing stage. We are not evaluating 50 random questions; the algorithm dynamically queries the 50 questions whose calibrated difficulties perfectly match the model's current estimated ability, maximizing the Fisher Information at each step. Previous work demonstrates that a small, targeted query budget yields the same statistical reliability as testing thousands of random items (Truong et al., 2025). The variance remains tightly bounded regardless of the total benchmark size $N$ because the effective information saturates quickly when relying only on maximally discriminative questions.
>
> We empirically report the MAE between CAT-estimated $\theta$ (using a limited budget) and ground-truth $\theta$ (from the full dataset) across all benchmarks, as shown in [Figure](https://anonymous.4open.science/r/irsl_rebuttal_figures-4205/pretrain_cat_budget_ablation.png) and [Figure](https://anonymous.4open.science/r/irsl_rebuttal_figures-4205/testtime_cat_budget_ablation.png) for pretraining and test time scenario, respectively, the error rapidly decreases and cconsistently stabilizes around a budget of 50 queries. We will include this empirical ablation in the Appendix of the revised version. Thank you again for helping us improve the paper.
>
> Truong, Sang, et al. "Reliable and efficient amortized model-based evaluation." ICML 25.

---

> > ### Author Rebuttal · Reviewer_XXm6 · 2026-04-02
> >
> > Since the paper’s current average score is already quite high, I will not raise my score.

---

> > > ### Author Response · Authors · 2026-04-06
> > >
> > > Thank you very much for your time and constructive suggestion. We sincerely appreciate your support!

---

### Decision · Program_Chairs · 2026-04-30

**Decision:**

Accept (regular)

**Comment:**

This paper proposes Beta-IRT, a fitting approach combined with Item Response Theory from Psychometrics, to derive the scaling curves of LLMs, which disentangles latent model ability from question characteristics to reduce computational costs during the pre-training, leading to more reliable pre-training and test-time performance estimation.

Reviewers consistently appreciated this work's innovative combination of IRT with scaling laws, strong practical value for efficient LLM evaluation, extensive experiments, and improved sample efficiency across pre-training and test-time settings.  After rebuttal, all reviewers acknowledged the concerns, such as comparisons between Beta-IRT and binary baselines, and the cost comparisons, have been fully addressed, and some reviewers explicitly support the acceptance of this work.

We encourage the authors to include all the responses in the revision and further revise the paper following the reviewers' suggestions.